# Advancing the estimation of future climate impacts within the United States

Corinne Hartin[1], Erin E. McDuffie[1], Karen Noiva[3], Marcus Sarofim[1], Bryan Parthum[2], Jeremy Martinich[1], Sarah Barr[1], Jim Neumann[3], Jacqueline Willwerth[3], and Allen Fawcett[1]

[1]Cimate Change Division, Office of Atmospheric Protection, U.S. Environmental Protection Agency, Washington, DC, USA

[2]National Center for Environmental Economics, Office of Policy, U.S. Environmental Protection Agency, Washington, DC, USA

[3]Industrial Economics incorporated 2067 Massachusetts Ave, Cambridge, MA 02140

*Correspondence to*: Corinne Hartin (hartin.corinne@epa.gov)

**Abstract**

Evidence of the physical and economic impacts of climate change is a critical input to policy development and decision making. In addition to the magnitude of potential impacts, detailed estimates of where, when, and to whom those damages may occur, the types of impacts that will be most damaging, uncertainties in these damages, and the ability of adaptation to reduce potential risks are all interconnected and important considerations. This study utilizes the reduced-complexity model, the Framework for Evaluating Damages and Impacts (FrEDI), to rapidly project economic and physical impacts of climate change across 10,000 future scenarios for multiple impact sectors, regions, and populations within the contiguous United States (U.S.). Results from FrEDI show that net national damages increase overtime, with mean climate-driven damages estimated to reach $2.9 trillion USD (95% CI: $510 billion to $12 trillion) annually by 2090. Detailed FrEDI results show that of the analysed sectors, the majority of annual long-term (e.g., 2090) damages are associated with climate change impacts to human health, including mortality attributable to climate-driven changes in temperature and air pollution ($O_3$ and $PM_{2.5}$) exposure. Regional results also show that annual long-term climate-driven damages vary geographically. The Southeast is projected to experience the largest annual damages per capita (mean: $9,300 per person annually, 95% CI: $1,800-$37,000 per person annually), whereas the smallest damages per capita are expected in the Southwest (mean: $6,300 per person annually, 95% CI: $840-$27,000 per person annually). Climate change impacts may also broaden existing societal inequalities, with, for example, Black or African Americans disproportionately affected by additional premature mortality from changes in air quality. Lastly, we extend FrEDI projections are extended through 2300 to estimate the net present climate-driven damages within U.S. borders from marginal changes in greenhouse gas emissions. Combined, this analysis provides the most detailed illustration to date of the distribution of climate change impacts within U.S. borders.

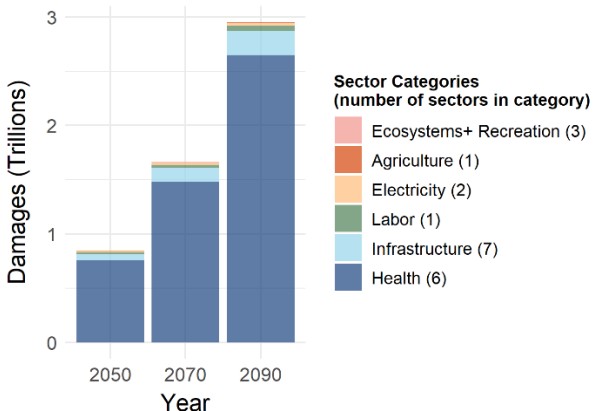

**Annual U.S. Climate-Driven Damages**

Mean Damages by Year and Category (Trillions $)

**Graphical abstract figure.**

**1 Introduction**

Evidence of the physical and economic impacts of climate change is a critical input to policy development and decision making. Information on the potential magnitude of climate change damages, where, when, and to whom those damages may occur, the types of impacts that will be most damaging, and the potential for adaptation to

reduce potential risks are all important and interconnected (Martinich et al., 2018). Understanding this rich set of information can help federal decision makers identify significant climate risks, which is as an important first step toward prioritizing and managing such risks, especially through mitigation and adaptation actions (GAO, 2017). Specifically in the U.S., results of recent multi-sector impact analyses show complex patterns of projected climate-driven changes across the country, with annual damages in some impact sectors (for example, labor, temperature -

related mortality, and coastal property) estimated to range in the hundreds of billions of U.S. dollars by the end of the century (Martinich and Crimmins, 2019; Hsiang et al., 2017).

Climate economics research has also continued to leverage recent advancements to develop and improve our understanding of damage functions that represent climate-driven impacts in broader economic frameworks (NAS, 2017). For example, advances in our understanding of the historical relationships between climatic variables and

the economy have enabled the development of methods to assess the economic effects from future climate change within the U.S. (GAO 2017; Field et al., 2014). As one example, the Climate Change Impacts and Risk Analysis (CIRA) project, coordinated by the USEPA and involving researchers from government, academia, and the private sector, has used and continues to use detailed sectoral models to quantify the physical and economic climate-driven damages across individual impact sectors within the U.S. (e.g., human health, infrastructure, and

water resources) (EPA, 2017a). Another example is the Climate Impact Lab -  a collaboration of more than 30 climate scientists, economists, and researchers from across the U.S. - which has focused its work on understanding the economic damages from climate change both within the U.S. (Hsiang et al., 2017) and across the globe,

including impacts to human health (Carleton et al., 2022), agriculture (Rising and Devineni, 2020; Hultgren et al., 2022), coastal property (Depsky et al., 2022), and energy (Rode et al., 2021).

Typically, these resource-intensive, bottom-up impact studies rely on a select number of large-scale global emission and warming scenarios (e.g., the Representative Concentration Pathways), limiting their ability to explore certain aspects of uncertainty associated with a wider range of alternative future trajectories. As an alternative approach, the Framework for Evaluating Damages and Impacts (FrEDI) (EPA, 2021b) draws upon information from these detailed sectoral impact studies to rapidly assess U.S. economic and physical impacts of climate change

within a common framework. FrEDI was developed using a transparent process, peer-reviewed methodologies, and is designed to be a flexible framework that is continually refined to incorporate advances in peer-reviewed economic damage functions, including the incorporation of new sectors and adaptation options. In this analysis, FrEDI draws upon over 30 climate change impact models from peer-reviewed studies to develop relationships between mean surface temperature change and climate-driven impacts across 20 sectors within U.S. borders,

through the end of the 21st century. FrEDI has the flexibility to use any custom warming scenario (which can be derived from a climate model e.g., Figure 1)) and couple it with ~~any~~ accompanying socioeconomic projections (e.g., gross domestic product (GDP) and population). Due to this level of detail and flexibility, FrEDI provides an efficient and transparent damage estimation approach to explore a variety of future baseline trajectories or emission reduction policies and, and thereby, can provide policy-relevant information and complement the types of

analyses and outputs provided by existing integrated assessment models.

In this study, we use 10,000 recently developed, paired probabilistic emissions and socioeconomic projections, in combination with resulting temperature projections from ~~with~~ a simple climate model ~~to provide~~ as inputs ~~for~~ to FrEDI, which ~~then~~ is then run to quantif~~ies~~ the annual physical and economic impacts associated with each~~of projected~~ resulting paired climate and socioeconomic ~~change~~ scenario through the end of the 21st century across

the contiguous United States (CONUS). This framework allows us to investigate the potential range of projected long-term annual climate change impacts that are associated with uncertainty in climate model parameters, a wide range of future emissions and socioeconomic conditions, as well as structural uncertainty in select damage functions. We present annual damages overtime and discuss the differential impacts projected to occur across different sectors, regions, and populations within CONUS borders to illustrate the breadth of the potential climate

change risks to the U.S. Lastly, we extend our methodology out to the year 2300 to assess the net present damage in the U.S. resulting from an additional ton of $CO_2$, $CH_4$ or $N_2O$ emissions. Aggregating net present damages across all sectors and regions within FrEDI provides a traceable estimate of the economic damages within U.S. borders, from a marginal change in greenhouse gas emissions.

**2 Methods**

This analysis consists of three components, each representing recent scientific advances in their respective fields (Figure 1). First, projections of global greenhouse gas emissions (Figure 1, Input 1) are used as input to a simple climate model to derive trajectories of changes in global mean surface temperature (Figure 1, Output 1). These emission projections were developed as paired scenarios with projections of national-level population and GDP, and therefore the resulting ~~These~~ temperature trajectories from the simple climate model are then passed to

FrEDI (Figure 1, Input 2) alongside the paired projections of U.S. Population and GDP (Figure 1, Input 1) to model annual long-term climate damages across 20 impact sectors, seven CONUS regions, multiple adaptation scenarios, and socially vulnerable populations (Figure 1, Output 2).

Specifically, we use 10,000 randomly sampled scenarios of global greenhouse gas emissions ($CO_2$, $CH_4$ and $N_2O$), U.S. population, and U.S. GDP from the Resources for the Future – Socioeconomic Projections (RFF-SPs) (Rennert

et al., 2021) (Section 2.1). Emission trajectories are input to the Finite Amplitude Impulse Response (FaIR) model, a simple emissions-based climate model (v1.6.2) that relates emissions to changes in global mean surface temperature (relative to 1850-1900 average) (Smith et al., 2018).~~, calibrated based on historical data and~~ ~~Intergovernmental Panel on Climate Change (IPCC) AR6 assessed climate variables (Smith et al., 2018).~~ The FaIR calibration is consistent with the IPCC AR6 Working Group 1 assessment of present-day warming, equilibrium

climate sensitivity, transient climate response, present-day aerosol radiative forcing, present-day $CO_2$ concentrations, and recent-past ocean heat content change, including the uncertainties in these distributions (Forster et al. 2021; Smith et al. 2021). The resulting 10,000 global mean surface temperature projections, along with corresponding population and GDP projections from the RFF-SPs, are then passed to FrEDI (v3.0) to calculate the physical and economic climate-driven damages. A unique feature of using probabilistic projections with a

simple climate model in this approach is the rich range of uncertainty parameters that can be assessed. However, there remain some limitations in that ~~Note that~~ separately considering climate parameter ~~uncertainty~~ ~~independently from~~and socioeconomic uncertainty ignores potential feedbacks from observed climate change onto socioeconomics (e.g., ~~if there is~~ a higher climate sensitivity~~, the~~could result in ~~resulting increase in~~larger climate-driven damages, which could lead to lower emissions or GDP than would occur in a lower climate

sensitivity world).

We describe each process in more detail below.

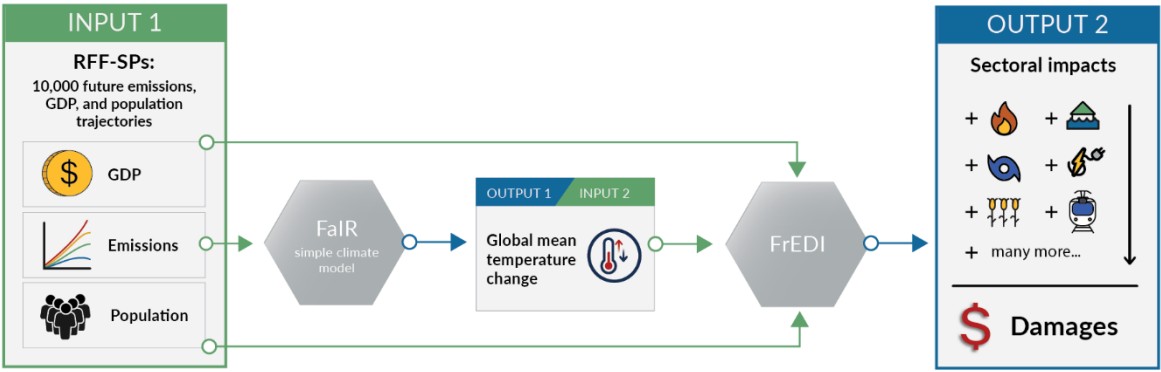

**Figure 1:** Flow diagram of the inputs and outputs needed to evaluate the economic damages within the U.S. Emission trajectories are passed as inputs into FaIR to calculate global mean surface temperature. Global mean surface temperature, population, and GDP are then passed as inputs to FrEDI to calculate sectoral climate impacts to the U.S. Not shown is the estimation of global mean sea level rise; these values are calculated within FrEDI using a semi-empirical approach from existing literature (Kopp et al., 2016) to calculate the impacts to the subset of FrEDI sectors that are impacted by sea level rise (i.e., transportation impacts from high tide flooding, and coastal properties) (EPA, 2021b).

### 2.1 Emissions and Socioeconomics

Socioeconomic and emissions projections from 2020 to 2300 were recently developed under the Resources for the Future Social Cost of Carbon Initiative (Rennert et al., 2021). These include multi-century probabilistic projections of country-level population, GDP, and global emissions of $CO_2$, $CH_4$ and $N_2O$. While uncertainties multi-century projections are considerable, as discussed in Rennert et al., 2021, tThese projections generally align with previous estimates and represent the largest a state-of-the-art set of probabilistic socioeconomic and emissions scenarios based on high-quality data, robust statistical techniques, and expert elicitation. In addition, tThese projections also incorporate coupled uncertainty in the time-dependent relationship between GDP and emissions, while also explicitly accounting for potential future climate policy and its contribution to the economy-emissions relationship (Rennert et al., 2021).

### 2.2 The Climate Model

The Finite Amplitude Impulse Response model (FaIRv1.6.2)[1] calculates atmospheric concentrations of greenhouse gases, radiative forcing, and global mean surface temperature from emissions of greenhouse gases, aerosols, and other gases (Smith et al., 2018). Version 1.6.2 was calibrated to and extensively used within the Sixth Assessment Report (AR6) of the IPCC (Forster, P. et al., 2021), resulting in 2,237 calibrated sets of climate parameters (out of the full 1 million member ensemble). While FaIR only captures uncertainties in those feedbacks and climate tipping points that are apparent in more sophisticated Earth system models or the historic record to which FaIR is calibrated, FaIR does include uncertainties in parameters such as the equilibrium climate sensitivity, transient

---

[1] https://github.com/OMS-NetZero/FAIR

climate response, present-day aerosol radiative forcing, present-day $CO_2$ concentrations, and recent-past ocean heat content change. Here we use the Monte Carlo simulation capabilities of MimiGIVE.jl (https://github.com/rffscghg/MimiGIVE.jl) to randomly sample the 10,000 RFF-SP emission scenarios (consisting of $CO_2$, $CH_4$, and $N_2O$) and the calibrated set of uncertain parameters contained in FaIR.[2] Emissions of the other gases and aerosols (e.g., HFCs, BC, OC, etc.) not included in the RFF-SP projections were set to the associated emissions in the SSP2-4.5 (Meinshausen et al., 2020) scenario, which most closely matches the median of the RFF-SP emission trajectories (Rennert et al., 2022). From the 10,000 model simulations, the average change in global mean surface temperature relative to 1986-2005 (FrEDI baseline) is 1.9°C (95% confidence interval: 0.8°C to 3.5°C) by 2100 and increases to 3.1°C (95% CI: -0.2°C to 7.8°C) by 2300 (Figure A1‑1).

**2.3 Damages from Climate Change to the U.S.**

The Framework for Evaluating Damages and Impacts (FrEDI) is a reduced complexity model that assesses and quantifies future impacts to the U.S. from a changing climate. As described in detail in the Technical Documentation (EPA, 2021b), FrEDI uses a temperature binning approach and data from previously published climate impact studies (Sarofim et al., 2021) to develop relationships between climate-driven changes in CONUS temperature or global mean sea level rise and the resulting physical and economic damages across 20 sectors (Table A2‑1) in seven U.S. regions. While FrEDI evaluates both negative and positive impacts of climate change across sectors and regions, climate-driven damages outweigh the positive effects for all sectors at the national level. FrEDI also provides insight into differences in impacts under various adaptation scenarios and contains a module that can be used to quantify impacts to socially vulnerable populations. The underlying studies in FrEDI consist of bottom-up detailed sectoral analyses from the CIRA project (EPA, 2017a) and other studies including those from the Climate Impact Lab (e.g., Hsiang et al., 2017) and the American Thoracic Society (e.g., Cromar et al., 2022). FrEDI was designed to fill the current need of monetizing a broad range of climate-driven impacts in the U.S. across various warming/emission/socioeconomic trajectories, while doing so in a significantly shorter computational timeframe (e.g., seconds) relative to existing impact models. To achieve this objective, the detailed spatial resolution of the underlying studies is reduced within FrEDI to the regional level, for ease, flexibility, and speed.

FrEDI currently includes 20 impact sectors for which damages are modelled as functions of a climate driver (CONUS temperature or sea-level rise), U.S. GDP, and regional population. The GDP and population projections from the RFF-SPs are at the country level (i.e., total U.S. population). For the analysis, we disaggregate national populations values from the RFF-SPs to populations for each of the seven FrEDI regions based on the percentage of regional to total U.S. population in the years 2010-2090 using projected regional populations derived from ICLUS (EPA, 2017b).

---

[2] See Rennert et al. (2022) for more detail on the RFF-SPs and FaIR parameter sets. Each of the 10,000 RFF-SPs are assumed equally likely.



Neither population projections, ICLUS or RFF-SPs, ~~The population projections from ICLUS~~ were generated considering future climate changes such as climate induced migration.~~: incorporating those feedbacks would lead to not accounting for costs incurred because of climate-induced migration.~~ The proportions for each region are held constant after 2090. Figure A1~~-1~~ shows that the mean and 95[th] confidence intervals for U.S. population and time-averaged U.S. GDP per capita growth rates are 390 million (95% CI: 260-520 million) and 1.5% (95% CI: -0.4% to 4.0%), respectively in 2100.[3] By 2300, the average of all 10,000 trajectories for U.S. population and time-averaged U.S. GDP per capita growth rates are 370 million (95% CI: 43 million to 1.3 billion) and 0.9% (95%CI: -0.2% to 3.4%), respectively. The trends shown in Figure A1~~-1~~ reflect the aggregate of the 10,000 individual RFF-SP trajectories (each of which has a different but~~,~~ equally likely growth path).




For sectoral impacts driven by temperature change, damages in FrEDI are calculated as functions of CONUS degrees of warming over time, relative to a 1986-2005 average temperature baseline. In this analysis, CONUS mean temperature change is estimated for each FaIR-derived temperature projection (calculated from each RFF-SP emissions scenario), as CONUS temperature (°C) =1.42 × Global Temperature (°C) (EPA, 2021b). This relationship between CONUS and global temperatures is relatively stable across GCMs and over time, allowing the use of these available datapoints to develop a generalized relationship between global and CONUS temperature anomalies. Sub-national differences in warming are also explored within FrEDI using results derived from a consistent set of GCMs that were also used within the underlying studies (e.g., Sarofim et al., 2021). For example, unique damage functions for each sector (and variant within each sector) are developed for each region and GCM, based on its relationship to CONUS temperature. While FrEDI outputs damages by region and GCM, the main results in this analysis present national and regional damages calculated from the average across the GCM ensemble. For sectoral impacts driven by sea level rise (i.e., coastal properties and transportation impacts from high tide flooding), global mean sea level is calculated within FrEDI from global mean surface temperature using a semi-empirical method that estimates global sea level change based on a statistical synthesis of a global database of regional sea-level reconstructions from Kopp et al. (2016). In FrEDI, for a given year, sea level-driven damages are calculated by interpolating between modelled damages at different sea level heights at that same point in time; this enables FrEDI to account for interactions between adaptation costs, increased coastal property values, and sea level rise over time (EPA, 2021b).


This analysis groups mean damages from each of 20 FrEDI sectors into six topical categories and uses the default FrEDI adaptation assumptions of 'Reactive', 'Reasonably Anticipated Adaptation', or 'No Additional Adaptation' (see Table A3~~-1~~) for each sector. As discussed further in Section A3, Reactive or Reasonably Anticipated Adaptation

---

[3] All dollar values in this paper are presented in 2020 U.S. dollars. Any necessary transformations in the inputs (e.g., RFF-SPs are in 2011$, ~~and~~ FrEDI takes in 2015$, and FrEDI results are presented in 2020$) are performed using the U.S. Bureau of Economic Analysis national data on annual implicit price deflators for U.S. GDP, the top row of BEA Table 1.1.9.

is where decision makers respond to climate change impacts by repairing damaged infrastructure (e.g., road or rail repair) or reactively responding to current conditions (e.g., building sea walls or beach nourishment), but do not

take actions to prevent or mitigate future climate change impacts. No ~~A~~additional ~~A~~adaptation largely incorporates historical or current levels of adaptive mitigation that were in place during the time period of each underlying sectoral study. Example sensitivities to projected climate-driven damages ~~A few adaption options~~ are explored within section 3.1 and A3.

FrEDI also has the capability to investigate adaptation options in select sectors. Available adaptation options reflect

the treatment of adaptation in the underlying sectoral studies. For most of these studies, because the implicit or explicit impact response functions are calibrated to historical or current data, historically practiced adaptation or hazard avoidance actions are "baked in", while enhanced adaptation action or new (currently unknown) technologies are not considered. Exceptions include FrEDI's coastal property and select other infrastructure sectors (e.g., roads, rail), where adaptation options and scenarios from the underlying studies have been

incorporated into FrEDI. Total damages in these sectors are sensitive to adaptation assumptions indicating that adaptation has the capacity to both exacerbate and ameliorate future climate-driven damages, with the latter being more common. These results are further explored below and in Section A3.

In addition to quantifying differential climate-driven damages across impact sectors, geographic regions, and adaptation options, FrEDI can also compare climate-driven damages across different populations within the U.S.

This capability is ~~largely~~ based on a recent EPA Report on Climate Change and Social Vulnerability in the United States (EPA, 2021a), which considers differential climate change risk as a function of exposure to where climate change impacts are projected to occur. ~~FrEDI incorporates this approach by using data on where populations live (US Census, 2014) as an indicator of exposure, and for vulnerability, considers four categories for which there is evidence of differential vulnerability (Table A2-2), including low income, ethnicity and race[4], educational~~

~~attainment, and age.~~ These differential impacts are calculated in FrEDI at the Census tract level as a function of current population demographic patterns (i.e., percent of each group living in each census tract) (U.S. Census), projections of CONUS population (U.S. EPA, 2017), and projections of where climate-driven damages are projected to occur (from Census tract-level temperature-impact relationships in FrEDI). The relative percent of each group in each Census tract is from the 2014-2018 U.S. Census American Community Survey dataset (U.S. Census) and is held

constant over time because robust and long-term projections for local changes in demographics out to 2090 and

---

~~[4] This analysis uses the term BIPOC to refer to individuals identifying as Black or African American; American Indian or Alaska Native; Asian; Native Hawaiian or Other Pacific Islander; and/or Hispanic or Latino. It is acknowledged that there is no 'one size fits all' language when it comes to talking about race and ethnicity, and that no one term is going to be embraced by every member of a population or community. The use of BIPOC is intended to reinforce the fact that not all people of color have the same experience and cultural identity. This report therefore includes, where possible, results for individual racial and ethnic groups.~~

beyond are not readily available. We consider ~~considers~~ four categories for which there is evidence of differential vulnerability (Table A2 ~~2~~), including low income, ethnicity, and race[5], educational attainment, and age.

**2.4 Estimating Net Present Value of Future Damages per ton of GHG Emissions**

While FrEDI was initially built to project damages through 2090 for temperature scenarios with a maximum value
of 10°C of warming,~~,~~ FrEDI was extended in this work to project climate damages out to 2300 to quantify the net present damages in the U.S. resulting from an additional tonne of $CO_2$, $CH_4$ or $N_2O$ emissions. As described further in Section A4, FrEDI is extended by linearly extrapolating its sector-specific, temperature-binned damage functions to account for the full range of temperature scenarios derived from the RFF-SP emission scenarios run through FaIR (some of which have degrees of warming above 10°C). To quantify the net present damages, all 10,000 RFF-
SP-derived temperature and socioeconomic scenarios are then run through FrEDI out to 2300 under two cases: a baseline (emissions = RFF-SP emissions) and a perturbed case, where 1 GtC pulse of $CO_2$ (or $CH_4$ or $N_2O$) is added to each of the RFF-SP emissions scenarios in the year 2020. The emissions are identical between the cases for all other years. The annual marginal climate-driven damages are calculated as the difference between the damages in the baseline and perturbed cases, summed across all sectors and all regions for each year. Lastly, these marginal
annual damages are discounted to the year of emissions and then aggregated across the timeseries into a single net present damage estimate. The results are normalized by the pulse size and gas chemistry (e.g., C to $CO_2$) and reported in 2020 U.S. dollars.

Future monetary impacts are generally discounted relative to present value. Circular A-4 (White House, 2003) recommends a constant value of 3% for the "social rate of time preference", which is considered to be the
appropriate discount rate to use for impacts on private consumption (which would include most environmental and health impacts). The discount rate of 3% was calibrated to the real rate of return for 10-year Treasury notes from 1973 through 2003. However, OMB Circular A-4 also noted that for intergenerational impacts (a category in which climate change clearly falls), discount rates lower than 3% might be appropriate. Moreover, recent real rates of return for Treasury notes have been lower than 3%, adding support for use of a discount rate smaller than 3%
(CEA, 2017). A number of economists, as well as the National Academies of Sciences (NAS, 2017) have alternatively suggested the use of Ramsey discounting (Eq. 2, $\rho$ is the rate of pure time preference, $g$ is a time-varying measure of per capita consumption or income, and $\eta$ is the elasticity of the marginal value of consumption with changes in $g_t$) as an appropriate approach to discounting long-term problems such as climate change. The effect of Ramsey

---

[5] This analysis uses the term BIPOC to refer to individuals identifying as Black or African American; American Indian or Alaska Native; Asian; Native Hawaiian or Other Pacific Islander; and/or Hispanic or Latino. It is acknowledged that there is no 'one size fits all' language when it comes to talking about race and ethnicity, and that no one term is going to be embraced by every member of a population or community. The use of BIPOC is intended to reinforce the fact that not all people of color have the same experience and cultural identity. This report therefore includes, where possible, results for individual racial and ethnic groups.

discounting is to value damages more highly in futures with less economic growth – e.g., future societies that have

fewer resources available for adaptation, and vice versa. A recent study from Rennert et al. (2022) used a Ramsey

approach calibrated to a near-term target discount rate of 2%, with $\rho$ = 0.2% and $\eta$ = 1.24.[6]  Here we use this

Ramsey discounting approach to calculate the net present value.

The net present value (NPV) for a constant discount rate ($r$) is calculated such that

$$NPV\big(D(t)\big) = \sum_{t=2020}^{t=2300} \frac{D(t)}{(1+r)^t} \qquad (1)$$

The net present value for a Ramsey discounting approach is calculated using a time-varying and state-specific

discount rate[7] which is a function of per capita economic growth ($gt$):

$$r_t = \rho + \eta * g_t \qquad (2)$$

and where this time varying rate is then used in the net present value calculation such that

$$NPV\big(D(t), g(t)\big) = \sum_{t=2020}^{t=2300} \frac{D(t)}{\prod_{x=2020}^{x=t}(1+r_x)} \qquad (3)$$

In this expression, $g_t$ has also been adjusted to reflect climate damages, such that in any given year $g_t$ is the per

capita consumption as calculated by taking the exogenous RFF-SP GDP, subtracting the damages output by FrEDI,

and dividing by total population. Because most of the sectoral damages ~~are~~ as determined from the underlying

sectoral models are proportional to GDP per capita (given that the default elasticity of VSL to GDP per capita is 1,

all sectors with a mortality endpoint also qualify), a correction can be made to account for this relationship

(Nordhaus, 2017). For this analysis, we use the equation

$$D\big(t, g(t)\big) = \frac{D_0(t)}{1 + D_0(t)\big/GDP_0(t)} \qquad (4)$$

Where $GDP_0(t)$ is the exogenous RFF-SP GDP, $D_0(t)$ is the initial total damages output by FrEDI, and D($t,g(t)$) are
the resulting damages.

**3 Results and Discussion**

**3.1 Annual U.S. Climate-Driven Damages by the End of 21st Century**

FrEDI was developed to quantify the physical and economic damages from climate change over the 21st century,

within contiguous U.S. borders. Figure 2 shows the net annual economic climate-driven damages across 20 sectors

---

[6] For Ramsey discounting calibrated to near-term target discount rates of 1.5%, 2.5%, or 3%, $\rho$ = 0.01%, 0.5%, and
0.8% and $\eta$ = 1.02, 1.42, and 1.57 respectively.
[7] Consistent with *Rennert et al. [2022]*, we use a stochastic Ramsey discount factor to discount future climate-
driven damages.

in the U.S. in the years 2050, 2070, and 2090, as calculated by the mean from the 10,000 baseline RFF-SP scenarios (i.e., emission, population, and GDP trajectories). Total annual damages throughout this analysis are shown in

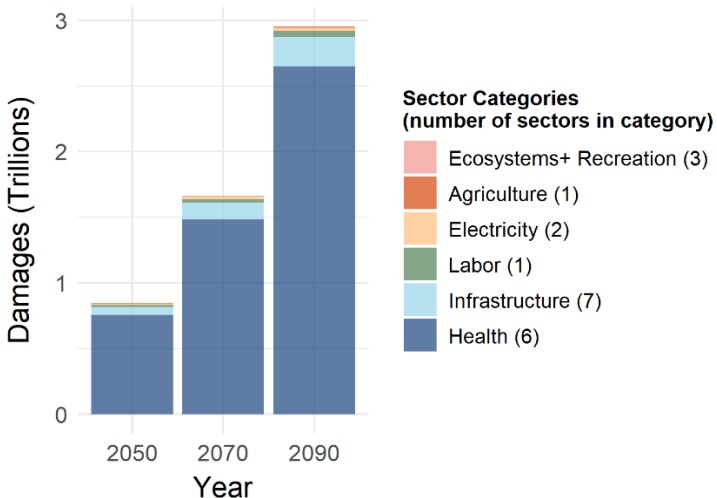

2020 U.S. dollars, converted from FrEDI's base units of $2015 USD using Annual GPD Implicit Price Deflators (U.S. Bureau of Economic Analysis, 2023). Figure 2 shows that net national damages increase overtime, with mean climate-driven damages estimated to reach $2.9 trillion USD (95% CI: $510 billion to $12 trillion), or ~3% of U.S. GDP, annually by 2090 for a subset of total climate impacts. Given that the drop in GDP in 2009 during the Great Recession was 2.2%[8], an annual decrease in GDP of over 3.0% per year by the end of the century (Figure 3) reflects

substantial damage to the national economy (though it is relevant to recognize that much of the damages estimated in FrEDI are a result of mortality, which is not directly reflected in historical GDP estimates). Table 1 provides the 2090 annual mean damages and 95% confidence interval (CI) for each aggregate category. Confidence intervals presented throughout this section include uncertainty in GDP, population, and climate parameters, but do not account for additional sectoral parametric or structural uncertainty. The individual sectors that contribute

to each category are listed in Table A2-1.

**Figure 2:** Annual mean U.S. climate-driven damages in 2050, 2070, and 2090. Damages are average values in billions of dollar (2020 USD) calculated from the 10,000 RFF-SPs. Sectors are grouped into six categories for visual purposes. The number of sectors included in each category is given in parenthesis in the legend. See Table A2-1 for the list of sectors in each cateogry.  Note that
this is only a subset of potential climate impacts to the U.S.

---

[8] Data from https://fred.stlouisfed.org/series/FYGDP, percentage decline in annual GDP from 2008 to 2009.

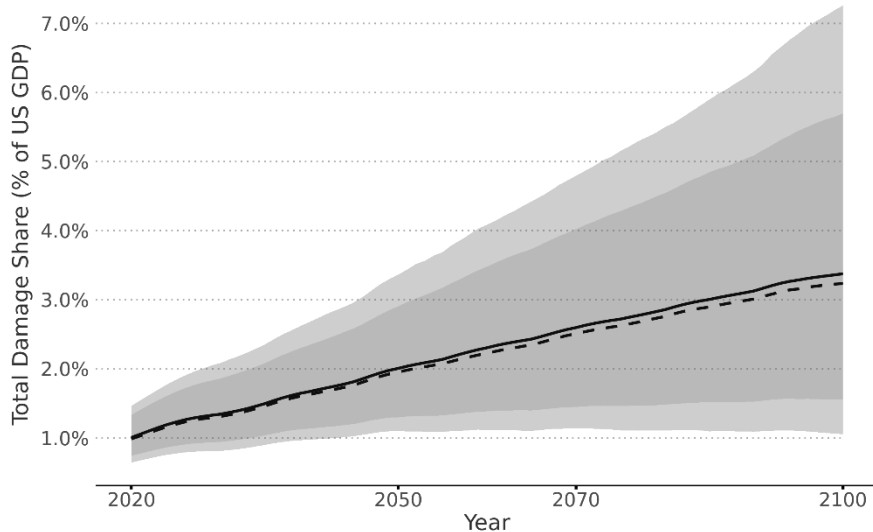

**Figure 3:** Share of U.S. GDP (from the RFF-SPs) of climate-driven damages for those impacts represented in FrEDI. Mean (solid) and median (dashed) lines along with 5th-95th (dark shaded) and 1st-99th (light shaded) percentile bounds.

**Table 1: The 95% confidence interval (CI) and mean annual U.S. climate-driven damages in 2090 for the six categories shown in Figure 2. All values are in 2020 USD. Totals may not sum due to rounding.**

| Category | Mean (billions) | 95% CI (billions) |
|---|---|---|
| Health | $2,600 | $350-$11,000 |
| Infrastructure | $220 | $140-$360 |
| Labor | $51 | $6.7-$220 |
| Electricity | $22 | $9.3-$35 |
| Agriculture | $6.1 | $0.42-$19 |
| Ecosystems + Recreation | $4.0 | $1.6 - $7.5 |
| Total in FrEDI | $2,700 | $510 – $12,000 |

Climate-driven damages from FrEDI are largest for the health category. The majority of damages in this category are from the estimated valuation of premature mortality attributable to climate-driven changes in temperature and air quality ($O_3$ and $PM_{2.5}$), but also include monetized health damages attributable to Valley fever, southwest dust, wildfire smoke exposure and suppression costs, and crime incidents. Another FrEDI category that includes the monetized value of directly estimated physical impacts (rather than a direct modelled relationship between temperature and monetized damages) is labor, which is the third largest category in 2090 and represents the damages resulting from lost hours of work when temperatures are too hot for workers to work outdoors or in unconditioned workplaces (e.g., warehouses). Table 2 provides the mean physical impacts from each of the sectors in the health and labor categories in 2090, along with the 95% CI. As shown in Table 2, climate-driven changes in

temperature have the largest impact on premature mortality, resulting in nearly 50,000 additional deaths (95% CI: 19,00-91,000 deaths) annually by 2090, followed by climate-driven changes in air quality (5,100 deaths; 95% CI: 2,100-10,000 deaths) and exposure to wildfire smoke (1,100 deaths; 95% CI: 460-1,700 deaths).

**Table 2: The range of 2090 physical impact results across the 10,000 RFF-SP projections, including the 95% CI and mean. Totals may not sum due to rounding.**

| Sector | Impact | 95% CI | Mean |
|---|---|---|---|
| Temperature Related Mortality | | 19,000 – 91,000 | 50,000 |
| Air Quality | Premature Mortality (deaths) | 2,100 – 10,000 | 5,100 |
| Wildfire | | 460 – 1,700 | 1,100 |
| Southwest Dust | | 160 – 690 | 390 |
| Valley Fever | | 130 – 480 | 300 |
| Crime | Incidence (number of property and violent crimes) | -160 – 1,100 | 4,700 |
| Labor | Work Hours lost (millions of hours) | 170 – 830 | 430 |

To further illustrate the distribution of monetized damages across sectors, Figure 4 shows the range of 2090 annual climate-driven damages in each of the 20 sectors in FrEDI, across all 10,000 RFF-SP emission, GDP, and population scenarios, in decreasing order of sectoral mean damages. Figure 4 shows that national total damages in 2090 are primarily driven by the valuation of premature mortality attributable to climate-driven changes in temperature (mean: $2.3 trillion per year; 95% CI: $0.31 – $9.9 trillion per year). The next four sectors with the largest monetary climate-driven damages include premature mortality attributable to changes in air quality (mean: $240 billion per year 95% CI: $32-$1000 billion per year), transportation impacts associated with changes in high tide flooding (mean: $140 billion per year; 95% CI: $110-$200 billion per year), national labor hours lost (mean: $51 billion per year; 95% CI: $6.7-$220 billion per year), and health damages from wildfire smoke exposure and response costs from wildfire suppression (mean: $51 billion per year, 95% CI: $8.1-$220 billion per year). Climate-driven damages to coastal properties associated with changes in tropical storm frequency and wind strength (mean: $29 billion per year; 95% CI: $12-$49 billion per year), damages attributable to changes in rail (mean: $19 billion per year; 95% CI: $7.7-$45 billion per year) and road systems (mean: $17 billion per year; 95% CI: $6.6-$35 billion per year), health damages from changes in southwestern dust exposure (mean: $18 billion per year ; 95% CI: $2.5-$77 billion per year), and the health burden of change in Valley fever incidence (mean: $14 billion per year; 95% CI: $2.0-$58 billion per year) round out the top 10 sectors with the largest annual damages in 2090. Figure A2̶1̶ provides the mean and 95% confidence interval total damages for each sector over the entire 2020-2100

340     timeseries. The large distribution of damages in each individual sector is driven by large range of RFF-SP emissions, population, and GDP projections and the dependence of the valuation approach for each sector on these parameters (as described in EPA, 2021b).

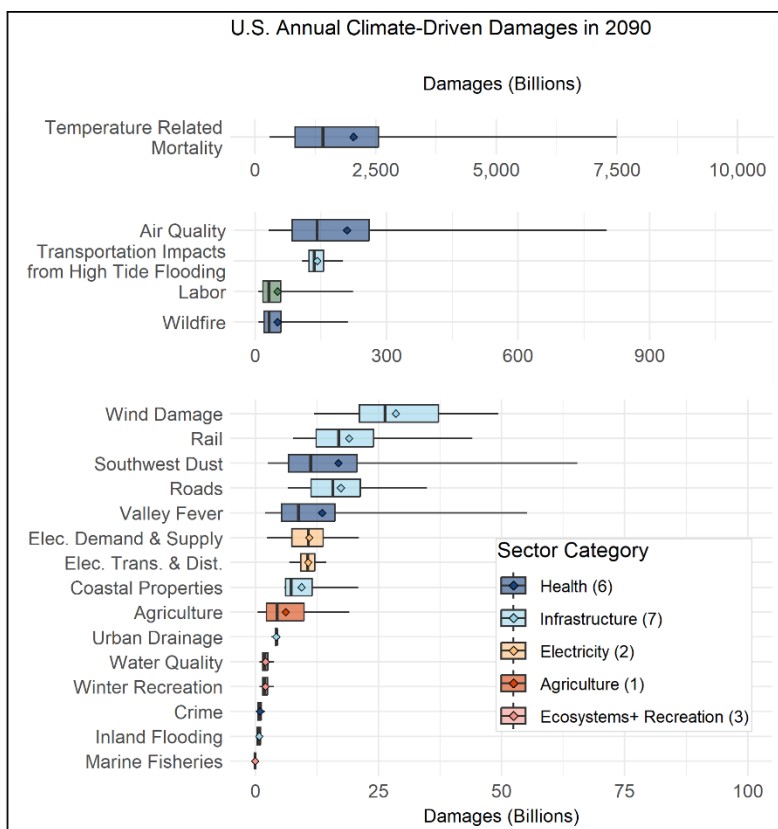

**Figure 4: Annual U.S. damages in the year 2090 by sector, in order of decreasing mean damages, colored by six sector category groupings. Note the change in x-axis in each panel. Box and whiskers show the 2.5th, 25th, 50th, 75th, and 97.5th percentiles, and mean damages (diamonds) across all 10,000 projections. Damages are in billions of 2020USD.**

These sectoral damages are sensitive to assumptions in the adaptation scenarios (see section A3 for more detail). For example, the coastal property sector considers three different adaptation options, no adaptation, reactive, and proactive adaptation. The underlying model within this sector, the National Coastal Property Model, has options

350     for ~~proactive retreat from the~~optimal ("proactive") response to future sea level rise~~shoreline, responsive~~"reactive" or reasonably anticipated response to current conditions (including sea walls, beach nourishment, house elevation, or managed retreat), ~~retreat~~ or ~~dealing with sea level rise and~~ rebuilding in place as often as necessary. Historical data suggests that most of our response to sea level rise thus far is in between ~~responsive retreat and dealing and rebuilding (i.e., no adaptation)~~reactive adaptation and no adaptation (Lorie et al., 2020)~~Lorie et al. 2020~~.

355     Considering the range of possible adaptation options in this coastal property sector, m~~M~~ean damages ~~for the coastal property sector~~ range from $17 billion USD under no adaptation to $7.5 billion USD under proactive

adaptation.  For this study ourDamages under the default 'reactive' adaptation assumption is reactive atare $9.4 billion USD. While the inclusion of adaptation options for any sector within FrEDI depends on the consideration and treatment of adaptation in the underlying impact studies, Table A3 further illustrates that projected climate-driven damages are sensitive to adaptation options in each sector where they are considered. Notably, the largest impact sector in this study, temperature-related mortality does not include assumptions about future adaptation. One important sector which does not include assumptions about future adaptation is temperature-related mortality: Wwhile the primary underlying study (Cromar et al., 2022) is a well-regarded meta-analysis of existing global temperature-related mortality studies, it does not explicitly consider future adaptive measures. Exploring projected 2090 damages from one alternative damage function that assesses impacts of extreme temperature on mortality in 49 U.S. cities (Mills et al., 2014), suggests that damages will be reduced (Table A4) in the event that U.S. cities can gradually adapt to hotter temperatures, for example through physical acclimatation, increased air conditioning penetration, and behavioral changes. SSeveral other studies have also shown observed reductions in historical temperature-related vulnerability over time (Lay et al., 2021),: however, there is nolittle consensus regarding the most appropriate way to consider future adaptation in this sector, even though several methods have been applied (Sarofim, M.C. et al., 2016; Carleton et al., 2022; Heutel et al., 2021). Therefore, we use the most recently published meta-analysis for the central estimate in this analysis, but also present results from alternative assumptions and studies (Tables A3 and A4), further illustrating the unique advantage of the FrEDI framework of enabling direct comparisons across studies.

The sectors addressedassessed in this study are generally independent and therefore damages are additive across these sectors. One potential exception could be temperature-related mortality and the climate-air quality linkage, as most approaches to estimating temperature-related mortality are statistical rather than mechanistic, which could lead to double counting of some health effects between these two sectors. Specifically, ((Cromar et al., 2022)) note that it will be important to continue exploring potential synergies between the effects of temperature and air pollution to adequately capture the potential risk in compound climate events such as these.. These potential overlaps should be considered when adding new sectors. Conversely, there can also be compounding effects that the FrEDI analytical this approach does not account for: e.g., power outages due to increased summer electricity demand could exacerbate temperature-related mortality. However, few studies produce quantitative, monetized estimates of compounding or interacting effects at the national scale as would be required to build into an FrEDIcomprehensive impact tools (Clarke et al. 2018).

Results from FrEDI also show that climate-driven damages across the national population vary by geographical region. Figure 5 shows a map of the damages per capita in each CONUS region in the year 2090, with pie charts showing the per capita damages in each region and the share of the four sectors with the largest damages (same figure for absolute regional damages in Figure A32 2). Based on the climate impacts included in FrEDI, Figure 5 shows that the Southeast will experience the largest annual damages per capita (mean: $9,300 per person

annually, 95% CI: $1,800-$37,000 per person annually), whereas the smallest damages per capita are expected in the Southwest region (mean: $6,300 per person annually, 95% CI: $840-$27,000 per person annually). In each region, the largest monetary damages in 2090 are expected from premature mortality associated with changes in temperature, ranging from $4,500 per person in the Southwest to $6,500 per person in the Southeast. Damages

from transportation impacts from high tide flooding and premature mortality attributable to climate-driven change in air quality are the second and third largest in the coastal Southeast and Northeast regions. In the Northwest and Southwest, the sectors with the second and third largest climate-driven monetized damages are air quality and wildfires. In the Southern Plains, high tide flooding transportation impacts and labor hours lost are the second and third largest sectors, while rail and wildfires are the second and third largest in the Northern Plains, and labor and

rail are the second and third largest in the Midwest. There are some regions and sectors projected to benefit from warming temperatures, including an expected reduction in air pollution attributable mortality in the Midwest under warmer conditions. Overall, however, the negative impacts of climate change outweigh the positives such that net losses are projected in each region.

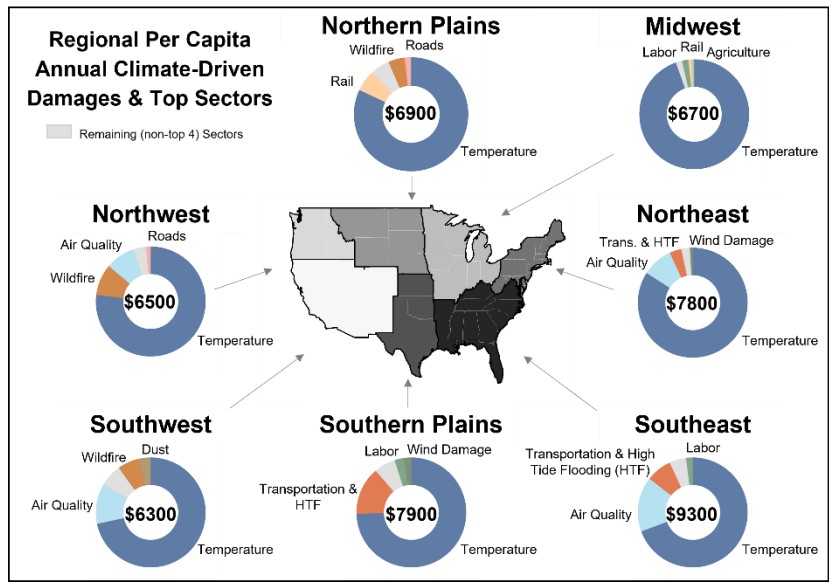

**Figure 5: Mean per capita annual climate-driven damages across the seven regions in 2090 for the subset of climate impacts included in FrEDI. Donut charts show the annual per capita damages (2020$ per person annually) and the top four sectors with the largest damages in each region. All damages from remaining (non-top four) sectors are shown by the light gray wedges.**

Lastly, climate change may also broaden existing societal inequalities (EPA, 2021a), and understanding the

comparative risks to different populations is critical for developing effective and equitable strategies for responding to climate change. As described in Section 2, FrEDI contains a module to generate and report results of disproportionate exposure and distributional physical effects across four groups of potentially socially vulnerable populations for six sectors. For example, results from this module show that Black or African Americans are more

likely to be affected by additional premature mortality from climate-driven changes in air quality, while Hispanic or

Latino Americans are more likely to experience lost labor hours (Figure 6) under a changing climate.

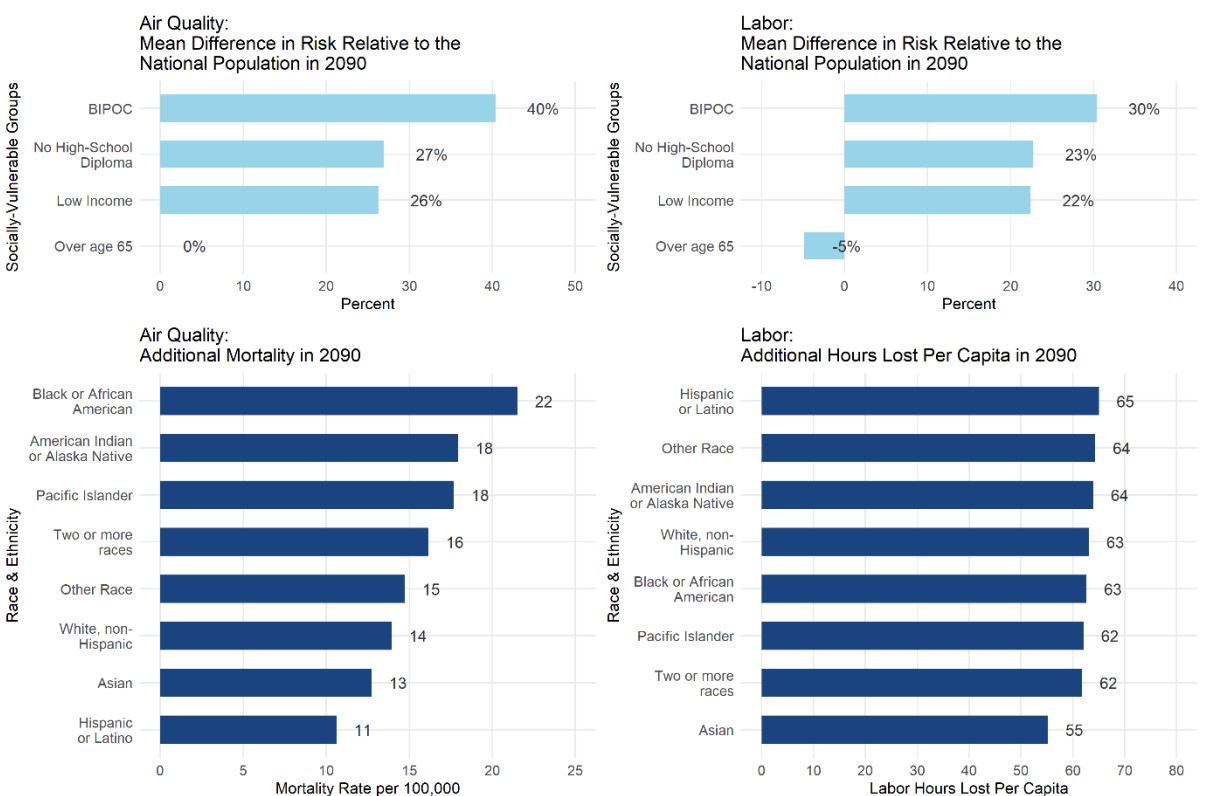

**Figure 6: Vulnerability to climate-driven changes in air quality attributable mortality and labor hours lost, by race and vulnerable groups in 2090. Top panels) Difference in risk in 2090 for four vulnerable populations. Bottom panels) Additional**
**rates of impacts in 2090, by race and ethnicity.**

Confidence intervals presented throughout this analysis account for uncertainty associated with the range of future emission and socioeconomic projections across the 10,000 RFF-SP scenarios. These also incorporate climate parameter uncertainty as a Monte Carlo approach was used to sample the calibrated parameter set when running

FaIR with the 10,000 RFF-SP emissions scenarios. In addition to these uncertainties and sensitivities to adaptation options, damage estimates within FrEDI are also sensitive to uncertainties in the underlying damage functions themselves. Similar to adaptation, FrEDI can incorporate parametric uncertainty in each damage function when the relevant information is available in the underlying study, as well as ~~this source of~~ structural uncertainty ~~when uncertainty estimates are available in the underlying study or~~ when multiple damages functions are available for a

single sector. For example, as further described in section A4, FrEDI incorporates three studies of climate-driven temperature-related mortality, two of which include underlying uncertainty estimates. As shown in Table A4~~3 2~~,

there is a large range of damage estimates from temperature-related mortality across each study, however, these values all fall within the uncertainty range derived from the RFF-SP scenarios, presented in the main text.

### 3.2 Comparison with SSPs

To place mean damages in context of alternative future storylines, Table 3 shows a comparison of annual national climate-driven damages in the U.S. in the year 2090 from a subset of four Shared Socioeconomic Pathways (SSPs), which represent projected socioeconomic global changes up to 2100 (O'Neill et al., 2017). Annual damages in Table 3 are calculated following the same approach as outlined in Figure 1, but using SSP trajectories of emissions, U.S. GDP, and U.S. population from the SSP Public Database (v2.0)[9]. These trajectories do not include uncertainty

related to climate and so we present only one value for each trajectory. Table 3 shows that annual U.S. climate-driven damages in 2090 from all but the SSP5-8.5 scenario fall below mean U.S. annual damages as predicted by the RFF-SP scenarios ($3.1 trillion). However, annual damages from all SSP scenarios fall within the 95% confidence interval ($0.5-$12.3 trillion).

**Table 3: Comparison of FrEDI damages from SSP and RFF socioeconomic input scenarios in 2090 (billions $2020 USD)**

| Scenario | Annual U.S. Damages (billion $2020USD) | Temperature Change in 2090 relative to FrEDI baseline (1986-2005 average) |
|---|---|---|
| SSP1-1.9 | 700 | 0.64 |
| SSP2-4.5 | 1700 | 1.8 |
| SSP3-7.0 | 1600 | 2.7 |
| SSP5-8.5 | 7000 | 3.4 |
| This study mean (95% CI) | 2900 (510-12,000) | 1.8 (0.80-3.2) |


### 3.3 Net Present Damages per ton of GHG emissions

We extend FrEDI to project climate damages through to 2300 (Section A4, Table A5) to quantify the net present damages within the US resulting from an additional tonne of $CO_2$, $CH_4$, or $N_2O$ emissions.[10] As described in Section 2.4, the net present value is the discounted sum of a stream of future damages produced by an emissions pulse in

2020 over the entire 2020-2300 time period. We explore the sensitivity of the remaining estimates to discounting assumptions by using Ramsey discounting calibrated to near-term target rates of 1.5%, 2.0%, 2.5%. Figure 7 shows the average, median, and range of estimated values for each discounting approach.[11]

---

[9] https://tntcat.iiasa.ac.at/SspDb/dsd?Action=htmlpage&page=80)
[10] Net present damages resulting from an additional ton of $CO_2$ emissions is sometimes characterized as a "domestic social cost of carbon."
[11] Figure A54-2 additionally compares these results to those using a constant discount rate of 3%, for a comparison with the historical approach in Circular A-4 (White House, 2003).

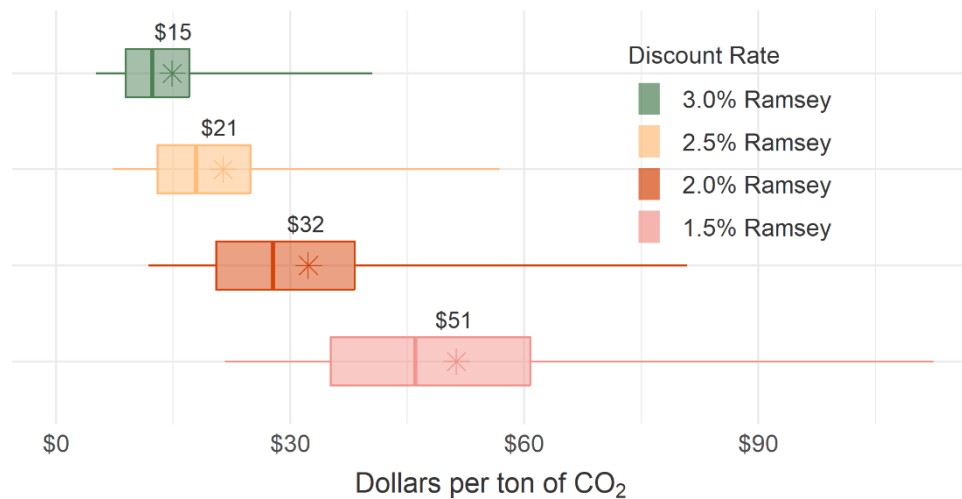

**Figure 7. Net present value of future damages from one tonne of CO$_2$ for damages occuring only within the CONUS. Units are in dollars (2020 USD) per ton of CO$_2$ emitted. Whiskers represent the 2.5$^{th}$ and 97.5$^{th}$ percentiles, while boxes span the 25$^{th}$ to 75$^{th}$. Mean values (stars and text) along with median values (vertical lines) are also shown.**

These results show that even considering only the direct CONUS impacts as estimated by FrEDI, damages per tonne of CO$_2$ are almost 20% of a recently estimated global value ($185 per tonne of CO$_2$ under a 2% Ramsey discounting, (Rennert et al. 2022)). This methodology can also be extended to explore the net present value of future damages resulting from an additional tonne of CH$_4$ (500$/ton of CH$_4$ under a 2% Ramsey discounting), N$_2$O (9,700$/ton of N$_2$O under a 2% Ramsey discounting), or other greenhouse gas emissions.

We recognize that multi-century projections are inherently challenging. This is particularly true for socioeconomic projections of GDP, population, and technologies: even projections to the end of the century have been challenged (Barron, 2018). The climate system is better understood, but FaIR only captures the effects of those feedbacks and tipping points which are apparent in the GCMs and historic record to which FaIR was calibrated.

While the damages estimated within FrEDI are constrained to the 48 contiguous United States, it is important to note that the appropriate climate damages to consider when evaluating policy-induced changes in a global pollutant such as greenhouse gases would be damages that account for impacts around the globe. For example, The National Academies of Sciences advised that "[i]t is important to consider what constitutes a domestic impact in the case of a global pollutant that could have international implications that affect the United States" (NAS, 2017). Impacts that occur outside of U.S. borders (and outside of FrEDI) will impact the welfare of U.S. residents and firms because of the interconnectedness of the global economy, international markets, trade, tourism, national security, political destabilization, additional spillover effects, and many other activities not yet captured in FrEDI. Moreover, the act of international reciprocity has been highlighted as motivation for including damages occurring outside of U.S. borders in a social cost estimate of global pollutants (Carleton and Greenstone, 2022; Revesz et al., 2017; and references within). It has also been shown that accounting for global damages in domestic policymaking can be individually rational (Kotchen, 2018). Therefore, we emphasize the contribution of the

damages estimated within FrEDI as providing a useful understanding of the channels through which climate change can affect U.S. citizens and residents and their relative magnitudes beyond what is currently possible in many

global models yet remain a partial estimate of the total damages from greenhouse gas emissions.

**4 Conclusions**

This study presents an evolving framework to quantify the damages of climate change to the U.S. economy, relying on more than a decade of research exploring individual sectoral impacts within the contiguous U.S. (EPA, 2021b). Impacts are dependent upon a change in global mean surface temperature, U.S. GDP and U.S. population, and

assumptions about adaptation. Adaptation is relevant in many sectors when quantifying benefits (Section A3), however, there are some sectors within FrEDI that do not have explicit options to model adaptation.  For example, the largest sector, premature mortality from temperature changes, dominates the monetized damages across all regions. The mortality approach used in this paper is based on a well-regarded systematic review and meta-analysis of temperature-related mortality studies (Cromar et al., 2022). However, there is substantial uncertainty

based both on difficulty of relating historical mortality to temperature changes, but also the potential for future adaptive responses to reduce vulnerability to temperatures (Carleton et al., 2022; Lay et al., 2021).

While this work advances our understanding of climate-related impacts to the U.S., it is far from a comprehensive accounting of sectoral damages within the U.S. The FrEDI framework is dynamic, with new sectors being added to the framework on a continuous basis (including in the near term several types of health impacts including mental

health, vibriosis, and health impacts of extreme storms), as well as broader coverage of direct and indirect impacts of inland flooding. However, the framework still omits coverage of many nonmarket sectors such as biodiversity, ocean acidification, many other ecosystem service losses, climate-forced migration, conflict, etc. We anticipate that the inclusion of more sectors will increase the estimates of net present damages due to GHG emissions. This work also omits the impacts of tipping elements due to climate change, which may lead to abrupt and irreversible

impacts (Armstrong McKay et al., 2022). This study does not explore tipping elements like permafrost thaw or Antarctic ice sheet instability. Future work may entail coupling BRICK to the framework to better explore the uncertainty within sea level rise (Wong et al., 2022, 2017) or coupling to an alternative reduced-form climate model, Hector, to explore permafrost thaw (Woodard et al., 2021).  Without explicit representation of some of these feedbacks, we can view these results as potentially lower bound damage estimates.  While $CO_2$ fertilization

effects are included in the damage estimates for the agriculture sector, the work does not account for any other direct effects of GHGs, such as the health, agriculture, or ecosystem damages resulting from ozone produced by methane's reaction in the atmosphere.  Lastly, this work does not account for interactions among sectors, interactions between non-U.S. and U.S. damages through global markets, and their feedback on the U.S. economy. While we focus on U.S. damages, we acknowledge that impacts resulting from GHG emissions, regardless of where

they originate, are global in nature. The bulk of the economic damages from climate change will be outside of the

U.S. and the U.S. may also experience indirect effects through trade, business, migration, etc. (NAS, 2017; Hsiang et al., 2017).

Regardless of these limitations, this work significantly advances our understanding of the impacts from climate change to the U.S., in what U.S. regions impacts are happening, what sectors are being impacted, and which
population groups being impacted the most. These results imply that there can be significant benefits to the U.S. from greenhouse gas mitigation, and significant benefits to the people of the U.S. FrEDI can also quantify the benefits of mitigation policies by comparing two scenarios similar to the results presented in section 3.3. Due to FrEDI's flexible framework, it allows for the model to be continually updated as studies of impacts to new sectors, or updates to outdated sectoral studies become available. Since this work incorporates multiple disciplines,
emission projections, climate modeling, impact modeling, and economic communities, it has the potential to be a useful tool in bridging the research gap between these communities and helping to address some of the omitted climate change risks currently within this field (Rising et al., 2022).

**Section A1: Detailed Inputs to FrEDI**

**Figure A1‑1.** Timeseries of global mean temperature (°C) relative to 1986-2005 baseline, U.S. population (millions), and average U.S. GDP percapita growth rate (2020$) for the 10,000 RFF-SP scenarios from 2020-2300. Temperature trajectories are derived from FaIR model runs of the 10,000 RFF-SP emission scenarios. Individual scenarios are shown by light gray lines. Medium and
dark gray shaded regions represent the 99[th] and 95[th] percent confidence intervals, respectively. The red line is the mean value overtime.

## Section A2: Detailed results to 2090

**Table A2-1. National Annual Damage Statistics (mean and 95% confidence interval) for the year 2090, in billions of 2020 USD, listed alphabetically by Sector**

| Sector | Category | Default Adaptation or Variant | Impact Type | 95% CI ($billion/year) | Mean ($billion/year) |
|---|---|---|---|---|---|
| Agriculture | Agriculture | With $CO_2$ fertilization | Revenue lost from changes in wheat, cotton, soybean, and maize crop yields | $0.42-$19 | $6.1 |
| Coastal Property | Infrastructure | Reactive Adaptation | Damage to coastal property value | $5.9-$21 | $9.4 |
| Electricity demand and supply | Electricity | No Additional Adaptation | Increases in power sector costs (e.g., capital, fuel, variable operation and maintenance (O&M), and fixed O&M cost) | $2.4-$21 | $11 |
| Electricity transmission and distribution | Electricity | Reactive Adaptation | Damages to transmission & distribution infrastructure | $6.9-$14 | $11 |
| Temperature-related mortality | Health | No Adaptation | Mortality from changes in hot and cold temperatures | $310 -$9,900 | $2300 |
| Hightide Flooding and Traffic | Infrastructure | Reasonably Anticipated Adaptation | Costs of traffic delays from flooding and cost of related infrastructure improvements | $110 -$200 | $141 |
| Inland Flooding (residential) | Infrastructure | No Additional Adaptation | Damages from riverine flooding | $0.1-$1.6 | $0.74 |
| Labor Allocation | Labor | No Additional Adaptation | Damages from work hours lost | $6.7-$220 | $51 |
| Marine Fisheries | Ecosystems + Recreation | No Additional Adaptation | Changes in thermally available habitat for commercial fish species | -$0.1-$0 | -$0.06 |
| Long-Term Air Quality Exposure | Health | 2011 Precursor Emissions | Mortality from ozone and fine particulate matter exposure | $32 -$9,900 | $230 |
| Property and Violent Crime | Health | No Additional Adaptation | Change in the number of Property and Violent crimes | $0.1-$2.0 | $0.92 |
| Rail Infrastructure | Infrastructure | Reactive Adaptation | Infrastructure costs associated with temperature-induced track buckling | $7.7-$45 | $19 |
| Road Infrastructure | Infrastructure | Reactive Adaptation | Cost of road repair, user costs (vehicle damage), and road delays due to changes in road surface quality | $6.6-$35 | $17 |
| Southwest Dust | Health | No Additional Adaptation | Mortality from changes in fine and coarse dust particle exposure | $2.5-$77 | $18 |
| Tropical Storm Wind Damage | Infrastructure | No Additional Adaptation | Cost of changes in hurricane wind damage to coastal properties | $12-$49 | $28 |
| Urban Drainage | Infrastructure | Proactive Adaptation | Costs of proactive urban drainage infrastructure adaptation | $3.2-$5.0 | $4.2 |
| Water Quality | Ecosystems + Recreation | No Additional Adaptation | Willingness to pay to avoid water quality changes | $0.83-$3.8 | $2.0 |
| Wildfire Air Quality Health Effects and Suppression Costs | Health | No Additional Adaptation | Mortality from wildfire emission exposure and response cost for fire suppression | $8.1-$210 | $51 |
| Winter Recreation | Ecosystems + Recreation | Adaptation | Revenue lost from suppliers of alpine, cross-country skiing, and snowmobiling | $0.83-$3.7 | $2.0 |
| Valley Fever | Health | No Additional Adaptation | Mortality, morbidity, and lost wages | $2.0-$58 | $14 |


Annual Climate-Driven Impacts

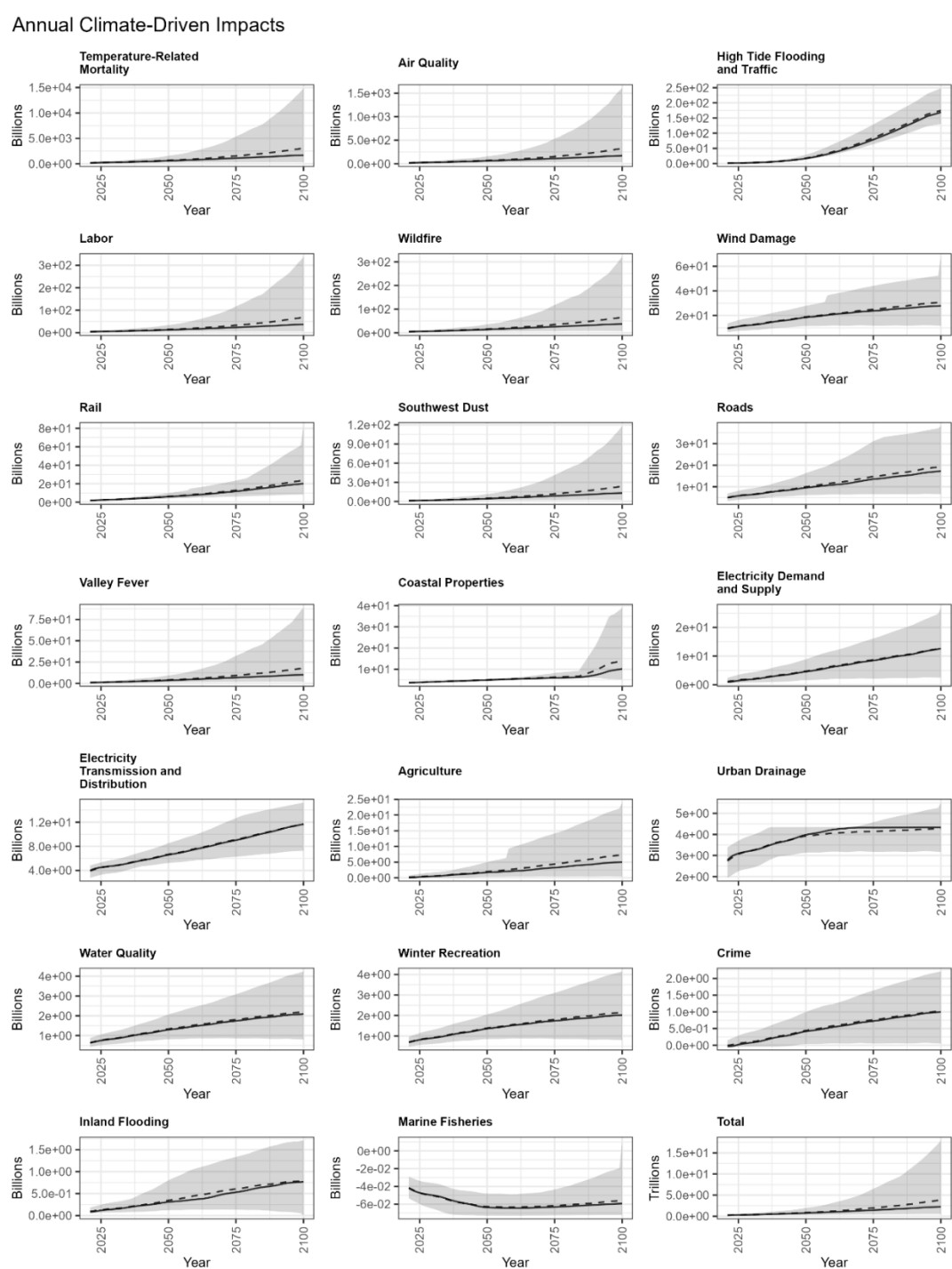

**Figure A2~~1~~.** Timeseries of sectoral damages in billions of 2020 USD across all 10,000 projections through 2100. Ordered by decreasing mean damages in the year 2090. Total damages (trillions) from all sectors in the lower right panel. Lines show annual mean (dashed) and median (solid) damages. Shaded areas show the 95% CI.. Temporal trends are a function of the underlying temeprature (or sea level rise) binned damage functions, as well as sector specific scalars (e.g., per capita income-dependent VSL).  Slight discontinuities in some of these sectors (e.g., agriculture) can occur either at the boundary between temperature bins (e.g., for agriculture and wind damage) or due to thresholds in the underlying studies. For example, the sharp increase in damages in the coastal property damage sector after 2080 are correspond to a sharp increase in damages that occur after sea levels breach 100 cm.


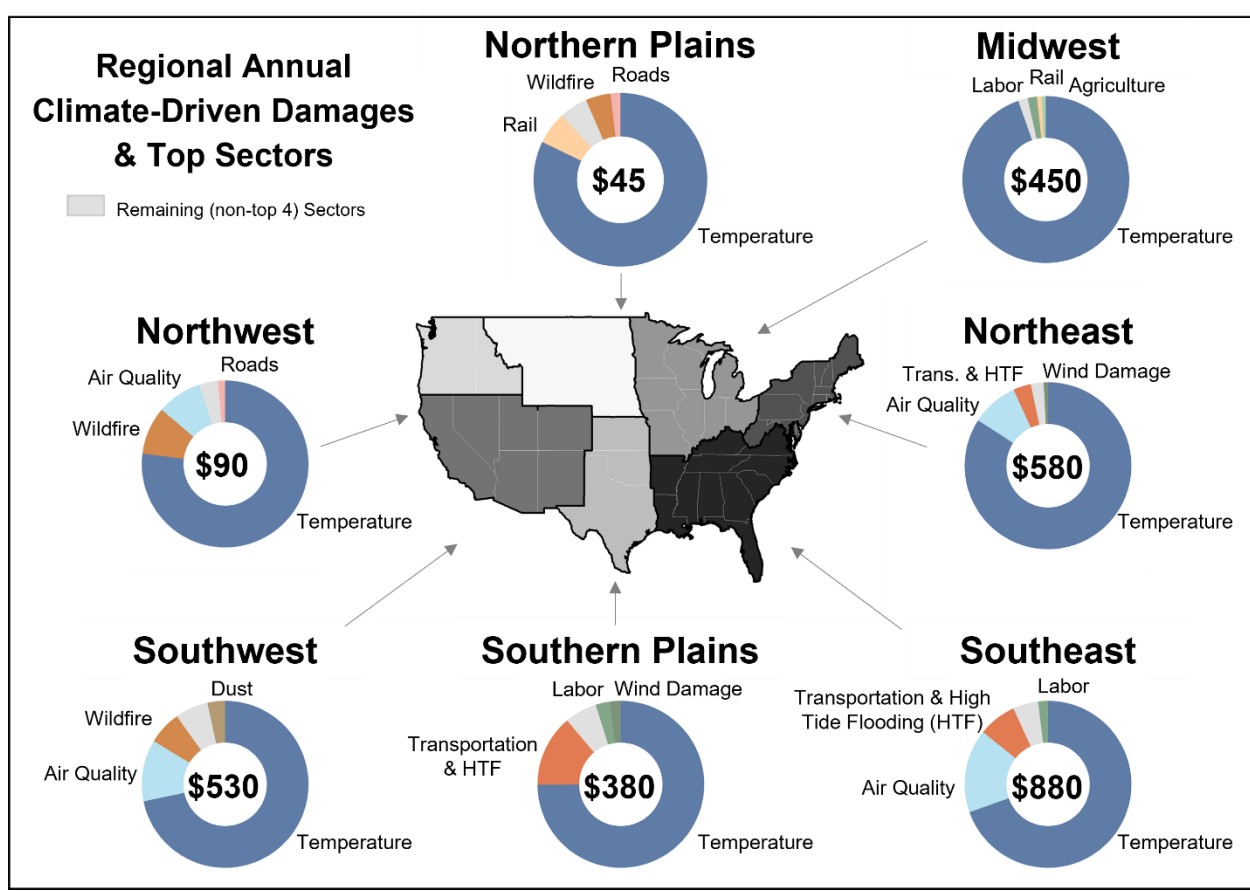


**Figure A32 2.** Map of mean annual climate-driven damages for a subset of sectors across 10,000 projections in each of the seven U.S. regions in the year 2090 (undiscounted). Damages are in billions of 2020USD. Donut charts show the absolute damages (in billions) in each region for those sectors included in FrEDI, and the top four sectors with the largest annual climate-driven damages. The share of damages from all remaining sectors are shown by the light gray wedge.


FrEDI also has a module to incorporate information from the recent EPA Report of Climate Change and Social Vulnerability in the United States: A Focus on Six Impacts (EPA, 2021a) (hereafter called the SV Report) to assess the differential climate-driven impacts in 2090 across different socially vulnerable groups. As described in the SV Report, this analysis considers four categories for which there is evidence of differential vulnerability. These groups

are listed in Table A2 2.


**Table A2 2.** Four socially vulnerable groups considered in this analysis and the reference groups (adapted from U.S. Environmental Protection Agency (2021a)

| Categories | Group Name | Description | Reference Group |
|---|---|---|---|
| *Income* | Low income | Individuals living in households with income that is 200% of the poverty level or lower | Individuals living in households with income greater than 200% of the poverty level. |
| *Age* | 65 and Older | Ages 65 and older | Under age 65 |
| *Race and ethnicity* | BIPOC | Individuals identifying as one or more of the following: Black or African American, American Indian or Alaska Native, Asian, Native Hawaiian or Other Pacific Islander, and/or Hispanic or Latino | Individuals identifying as White and/or non-Hispanic |
| *Education* | No High School Diploma | individuals aged 25 and older with less than a high school diploma or equivalent | Individuals aged 25 or older with educational attainment of a high school diploma (or equivalent) or higher. |

~~These differential impacts are calculated in FrEDI at the Census tract level as a function of current population demographic patterns (i.e., percent of each group living in each census tract), projections of CONUS population (from ILCUS, U.S. Environmental Protection Agency, 2017), and projections of where climate-driven impacts are projected to occur (i.e., using FrEDI temperature-impact relationships) at the Census tract level. The relative percent of each socially vulnerable group in each Census tract are from the 2014-2018 U.S. Census American Community Survey dataset (U.S. Census) and are held constant overtime because robust and long-term projections~~

~~for local changes in demographics are not readily available.~~

**Section A3: FrEDI Adaptation and Uncertainty Results**

FrEDI also has the additional capability to investigate some of these adaptation options in select sectors by reflecting the treatment in the underlying sector studies. FrEDI maintains adaptation assumptions from the underlying studies that form the basis of FrEDI's temperature-driven sectoral damage functions. For most of these studies, because the implicit or explicit impact response function is calibrated to historical or current data, this means that historically practiced adaptation or hazard avoidance actions are "baked in" – but enhanced adaptation action, or new (currently unknown) technologies are not considered. The exceptions include coastal property and select other infrastructure sectors, where the underlying studies consider specific adaptation actions. These have been incorporated into FrEDI.  For example, for the coastal flooding sector, FrEDI's default adaptation assumption is a Reactive Adaptation scenario, as defined in Neumann et al. (2021), and includes the costs (and reflects the hazard reduction benefits) to~~of~~ eleva~~t~~tion of~~r ion of~~ properties, and armoring where and when the benefits exceed the costs of this measure and expanded beach nourishment at locations where it is currently practiced. No other measures are included. There is an option in FrEDI, however, for the user to select either a No Adaptation scenario for this sector, which excludes the options above ~~to elevate properties~~ as well as measures that might hold back floodwaters, or a Proactive Adaptation scenario, where adaptation measures include elevation, beach nourishment, and armoring (either with bulkheads in protected areas or more expensive seawalls in areas exposed to higher open ocean wave action) and are chose based on the assumption that sea level will continue to rise in the future. It is difficult to comment on the realism of future action. There is some discussion in both Neumann et al. (2021) and Lorie et al. (2020), both of which make the point that even under current coastal hazards, cost-effective adaptation measures have not been adopted, probably because they involve short term capital investment to yield future, uncertain benefits. This is one reason why Proactive adaptation is not the default scenario in FrEDI.

For econometrically based sectors (e.g., Labor), adaptation is included to the extent that adaptation is currently occurring (e.g., work-place safety procedures currently being utilized to protect against extreme temperatures; individual risk/damage avoidance behavior reflected in current practice). For infrastructure sectors (i.e., Rail, Roads, Electricity Transmission and Distribution Infrastructure, Coastal Properties, and Transportation Impacts from High Tide Flooding), a no additional adaptation approach to infrastructure management does not incorporate climate change risks into the maintenance and repair decision-making process beyond baseline expectations and practice. The infrastructure sectors include two adaptation scenarios, following Melvin et al. (2017): Reactive adaptation, where decision makers respond to climate change impacts by repairing damaged infrastructure, but do not take actions to prevent or mitigate future climate change impacts (a variant on this scenario is the "Reasonably anticipated adaptation" option for the High-Tide Flooding and Traffic sector, which is defined similarly to the Reactive scenario); and Proactive adaptation, where decision makers take adaptive action with the goal of preventing infrastructure repair costs associated with future climate change impacts. This Proactive Adaptation

scenario assumes well-timed infrastructure investments, which may be overly optimistic given that such investments have oftentimes been delayed and underfunded in the past, and because decisionmakers and the public are typically not fully aware of potential climate risks (these barriers to realizing full deployment of cost-effective adaptation are described in Chambwera et al., 2014).

Table A3-~~1~~ shows that climate damages are sensitive to assumptions in the adaptation scenarios with mean 2090 annual damages of up to 2 to nearly 500 times larger in proactive or direct adaptation scenarios relative to damages when considering no adaptation. This illustrates adaptation has the capacity to both exacerbate and ameliorate future climate-driven damages.

**Table A3-~~1~~:** Annual mean (and 95% confidence interval) climate-driven damages in 2090 for sectors that include different adapation options. Damages are in billions of dollars (2020 USD).

| Sector | Adaptation Option [A] | Mean ($billions/year) | 95% CI ($billions/year) |
|---|---|---|---|
| Electricity Transmission and Distribution | No Adaptation | $12 | $7.3-$18 |
| | Reactive Adaptation | $11 | $6.9~~7.0~~-$14 |
| | Proactive Adaptation | $6.3 | $4.9~~5.0~~-$8.3 |
| Rail | No Adaptation | $21 | $7.2-$55 |
| | Reactive Adaptation | $19 | $7.7-$45 |
| | Proactive Adaptation | $1.5 | $0.28-$3.9 |
| Roads | No Adaptation | $130~~5~~ | $25-$330 |
| | Reactive Adaptation | $17 | $6.6-$35 |
| | Proactive Adaptation | $7.3 | $5.8-$8.4 |
| Coastal Properties | No Adaptation | $16~~7~~ | $~~10.1~~9.9-$37~~9~~ |
| | Reactive Adaptation | $9.4 | $5~~6.9~~6.90-$21 |
| | Proactive Adaptation | $7.5 | $7.0-$8.3 |
| Transportation Impacts form High Tide Flooding | No Adaptation | $890 | $680-$1,200 |
| | Reasonably Anticipated Adaptation | $140 | $110-$200 |
| | Direct Adaptation | $1.9~~2.0~~ | $1.3-$3.4 |

[A]Default adaptation assumption in FrEDI is the Reactive or Reasonably Anticipated Adaptation option

In addition to adaptation scenarios, FrEDI also has the capability to explore the sensitivity of future climate damages to specific changes in additional sectors, including agricultural damages with and without $CO_2$

fertilization, a lower air quality precursor emissions scenario, and high and low confidence intervals associated with damage functions~~s~~ ~~e~~specially from temperature-related mortality. The Cromar et al., 2022 study also provides a standard error on the impact function relative risk coefficient, which was used to develop a 90% ~~percent~~ confidence interval around this parameter. The 90% ~~percent~~ confidence interval supports the presentation~~calculation~~ of results~~impacts~~ for the low and high end of the confidence interval (5th and 95th percentile values) within FrEDI, as well as a central estimate which corresponds to the mean result. The Hsiang et

al., 2017 study authors also shared results from uncertainty modeling in the underlying work, which was also used to developed~~ed~~ a 90 ~~percent~~% confidence interval of results. These uncertainty ~~analysis~~ results support

presentationthe calculation of the low and high end of the confidence interval (5th and 95th percentile values) within FrEDI, as well as a central estimate which corresponds to the median result (50th percentile).

Within FrEDI, tThere are currently three underlying temperature-related mortality studies within FrEDI. Table A43-2 provides a snapshot of the parametric structural uncertainty within the each temperature-related mortality estimates, as well as structural damage function uncertainty by comparing impacts across multiple studies. Within FrEDI, there are currently three underlying temperature-related mortality studies. WeTo separately evaluate the level of damage-function-related uncertainty compared to other sources of uncertainty presented in the main text (e.g., socioeconomics & climate), we evaluate with show the mean damages from each damage function in Table

A4, as calculated as the average across from the RFF-SPs, as well as the 905th confidence intervals, as calculated by taking the average across the RFF-SPs for the damages predicted by the high and low confidence interval damage functions. from the mean RFF-SPs. These confidence intervals are taken from the uncertainty within the underlying study. Compared to Table A1, Table A4 shows smaller predicted ranges in temperature-related mortality damages than the ranges in damages derived from combined uncertainties in socioeconomic and climate

parameters. We do not present these uncertainty levels in the main text as only a select number of sectors currently included with the FrEDI framework include information that allow us to evaluate parametric and structural damage function uncertainty. We also note that the underlying data in Hsiang et al., 2017 is calculated as the median and therefore we are taking the mean across the RFF-SPs and the median damages. The Mills et al., 2014 study evaluates two scenarios, one with adaption and one without adaptation.

**Table A43-2.:** Annual mean (905th % confidence interval) climate-driven damages in 2090 for premautre mortality from temperature across three separate studies. Damages are in billions of dollars (2020 USD). Cromar et al., 2022 is used for temperature-related morltality throughout this the analysis presented in the main text.

| 2090 Temperature-Related Premature mortality – Billions 2020 USD | | |
|---|---|---|
| **Underlying Study** | **905th CI** | **Mean** |
| Cromar et al., 2022 | $300 - $3,900 | $2,100 |
| Hsiang et al., 2017 | $-280 - $1,800 | $740 |
| Mills et al., 2014 (w/ adaptation) | - | $31.0 |
| Mills et al., 2014 (w/o adaptation) | - | $110 |

**Section A4: FrEDI through 2300**

FrEDI was calibrated to estimate impacts for detailed 21st century scenarios and trajectories, as described in Sarofim et al. (2021). Extending the FrEDI approach to 2300 requires two adjustments to adapt the sensitivity of the model to climate drivers and to socioeconomic conditions beyond the 21st century. First, we consider how the sensitivity to climate drivers (temperatures and SLR inputs) might differ from 21st century conditions. FrEDI

damages were originally calibrated for temperatures from 0 to 6 degrees, relative to the 1986 to 2005 era, and SLR for 21st century trajectories that result in 30 to 250 cm GMSL outcomes by 2100. The original framework only returns physical and economic damage estimates within those bounds. In the modified FrEDI damage estimates for temperature inputs above these bounds are calculated by extrapolating damages per degree using the change in damages between 5 and 6 degrees. SLR inputs above the bounds are extrapolated based on the damages per

centimeter of SLR modelled by the two highest sea level scenarios in 2090.

Up to 6 degrees, FrEDI uses a piecewise linear function to estimate damages. This approach captures non-linearities from the underlying impact models. However, for temperatures above the calibration regime, FrEDI assumes a linear rate of change in damages equal to the change in damages from 5 degrees to 6 degrees. This assumptions is likely to be conservative: Hsiang et al. (2017) found that combined damages in the United States

increased quadratically with temperature, and ~~Weitzman~~ (Weitzman, 2012) suggested that while a quadratic damage form might be reasonable for temperature changes up to 2.5 degrees C globally, for higher temperatures it would make sense for damages to increase more quickly, as standard damage functions are unlikely to capture the sheer magnitude of impacts resulting from the kind of dramatic changes the planet would undergo at temperature changes substantially higher than that.

Second, we consider how the sensitivity to socioeconomic drivers continues beyond 2090 through 2300 on a sector-specific basis (Table A5~~4~~1). Damage estimates in FrEDI reflect year-specific socioeconomic conditions. There are several ways these conditions are defined through 2090 and linked to the damage estimates for temperature-based damages. Treatment for 2090 through 2300 is explained after the description of the original definition for each category of adjustments.

1.  **Impacts scale with population and/or GDP per capita.** For sectors with explicit links to population and GDP, temperature-based damage estimates are scaled based on the population and GDP trajectory for a defined run. This is most common for health sectors, where total cases scale linearly with population and valuation of cases scales with GDP per capita. For example, willingness to pay to reduce fatality risk (referred to as the value of statistical life or VSL) is adjusted based on the projection of GDP per capita and

680         a default income elasticity of 1.0. *2090 through 2300: Defined population and GDP trajectories continue to scale damage estimates through 2300.*

    2.  **Year-specific Adjustment Factors.** In sectors where population and/or GDP per capita enter the impact function in complex ways that cannot be extracted and replicated within the FrEDI framework, a series of

year-specific adjustment factors defined based on the underlying study are used to adjust damages over time and/or space. For example, changes in health outcomes over time driven by demographic composition (e.g., population by age group or geographic distribution within region, which affect baseline mortality rates or exposure) are incorporated in FrEDI as year-specific adjustment factors. These factors are derived from the underlying studies via two methods:

   a. By comparing per capita damage rates from a constant population run to a run that incorporates population growth[12], resulting in a time series of adjustment factors. *2090 through 2300: The time series of adjustment factors is either linearly extrapolated through 2300 or held constant at 2090 levels based on the observed trends 2010 through 2090 and the interpretation of the factor.*

   b. By comparing per capita damage rates for two constant population scenarios (i.e., 2010 and 2090) and interpolating for between years. *2090 through 2300: Per capita damage rate adjustments are held at 2090 levels through 2300.*

3. **No time-dependent adjustments.** Some sectors – which, in general, make up a small portion of overall damages– are not adjusted for socioeconomic projections but vary based only on sensitivity to projected temperature (Table A54 1). *2090 through 2300: No additional adjustments necessary.*

Some sectors utilize more than one method (e.g., southwest dust outcomes scale linearly with population, method 1 in list above, and per capita mortality rates are adjusted over time based on method 2a).

**Sea level rise-based damages** in FrEDI are derived from damages in the underlying studies that are year and sea level rise specific through 2100, thus no additional time-dependent adjustments are necessary for that timeframe. Damages in each year reflect real property prices and adaptation decisions made in previous periods. *2090 through 2300: Damages post-2100 are based on sea level rise-based damages from 2100 adjusted for real property price appreciation using GDP per capita and income elasticity of 0.45, consistent with the underlying Neumann et al., (2021).*

---

[12] Another, less common method for calculating adjustment factors is to compare two runs with and without climate change, each with population growth, to baseline damages (e.g. no population growth and no climate change).

**Table A5 4-1.** Summary of Strategy for Extending FrEDI Sectoral Results from 2090 to 2300 Modeling Horizon. Impact column provides detail for subcategories of impacts estimated within the Framework. Wildfire sector subcategories include morbidity and mortality associated with air quality impacts and fire suppression response costs – these two classes of subcategories are listed separately because they emloy different extension strategies.

| Sector | Impact | Extension Strategy |
|---|---|---|
| Air Quality | Ozone | Impacts continue to scale with population and/or GDP per capita *(Adjustment 1 in list above)* |
| | PM$_{2.5}$ | |
| Temperature-Related Mortality (Cromar et al., 2022) | N/A | |
| Labor | N/A | |
| Valley Fever | Mortality | |
| | Morbidity | |
| | Lost Wages | |
| Water Quality | N/A | |
| Wildfire | Morbidity | |
| | Mortality | |
| Winter Recreation | Alpine Skiing | Impacts continue to scale with population and/or GDP per capita *(Adjustment 1)* AND Year-specific adjustment factors developed from two constant population scenarios: per capita damages rates from 2090 applied 2090-2300 *(Adjustment 2b)* |
| | Cross-Country Skiing | |
| | Snowmobiling | |
| Southwest Dust | Acute Myocardial Infarction | |
| | All Cardiovascular | |
| | All Mortality | |
| | All Respiratory | |
| | Asthma ER | |
| Electricity Supply and Demand | N/A | Year-specific adjustment factors developed based on comparison of with and without population growth scenarios: extend existing scalars linearly past 2090 *(Adjustment 2a)* |
| Electricity Transmission and Distribution | N/A | |
| Roads | N/A | |
| Rail | N/A | |
| Coastal Properties | N/A | Sea level rise-based sectors: post-2090 impacts scale with GDP or GDP per capita |
| Transportation Impacts from High Tide Flooding | N/A | |
| Inland Flooding | N/A | No time dependent multipliers used to adjust temperature-driven impacts over time |
| Urban Drainage | N/A | |
| Wildfire | Response Costs | |
| Wind Damage | N/A | |
| Marine Fisheries | N/A | |
| Agriculture | Cotton | |
| | Maize | |
| | Soybean | |
| | Wheat | |
| Crime | Property | |
| | Violent | |

As described in the main text, FrEDI is run through 2300 (Figure A4 1) to calculate the net present damages associated with an additional pulse of 1 tonne of $CO_2$ in the year 2020. In addition to the Ramsey discounting approach presented in the main text, Figure A5 4 2 provides a comparison to the net present damages calculated

using a constant discount rate of 3%, consistent with OMB Circular A-4 (White House, 2003).

## Annual Climate-Driven Impacts

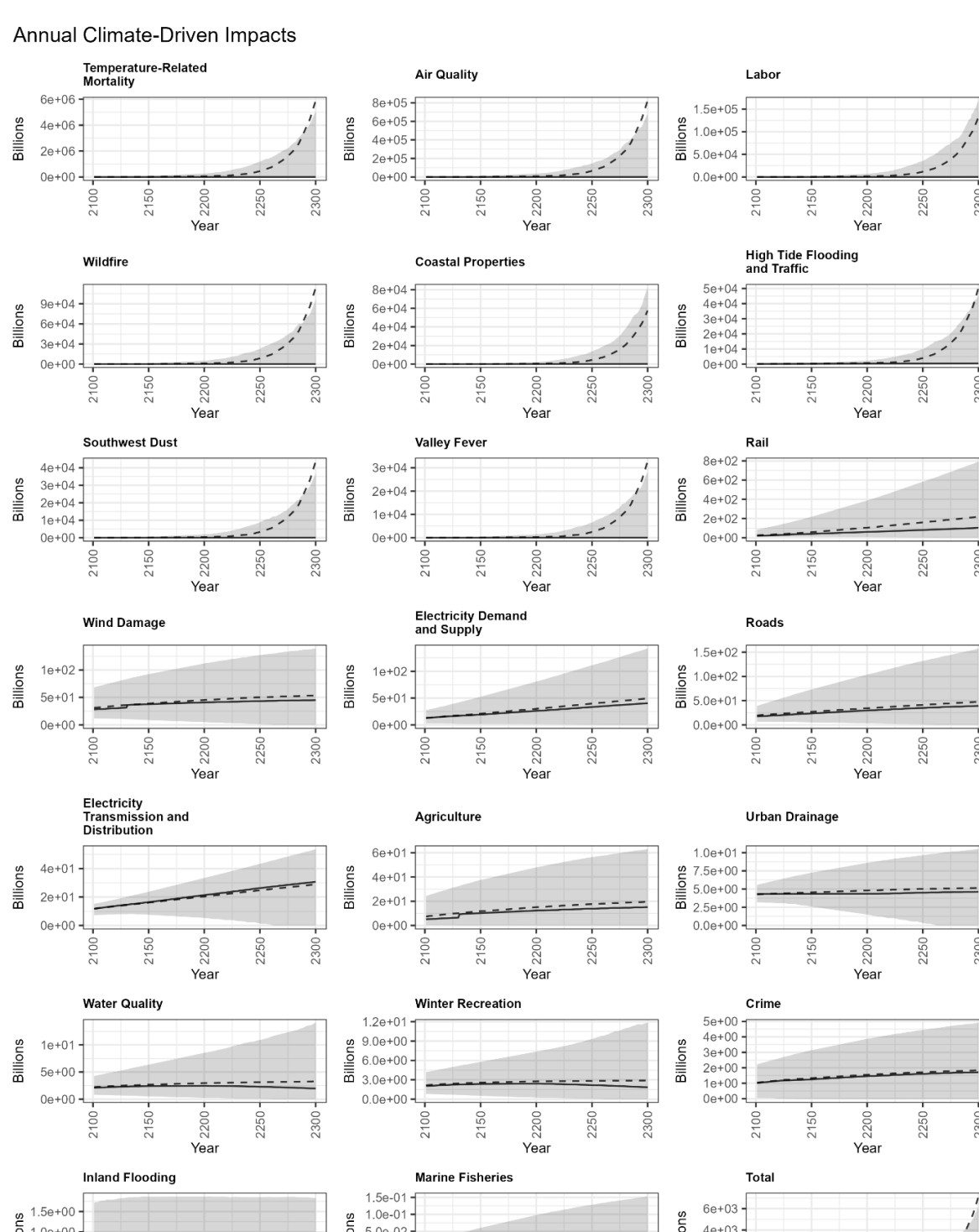

**Figure A4 1.** Timeseries of sectoral damages across all 10,000 projections from 2100-2300. Ordered by decreasing mean damages in the year 2300. The lower right panel shows total damages summed across all sectors. Dashed line (solid) shows the mean (median) damages each year. Shaded areas show the 95% CI. Annual damages are in units of billions of 2020USD (trillions for total panel).

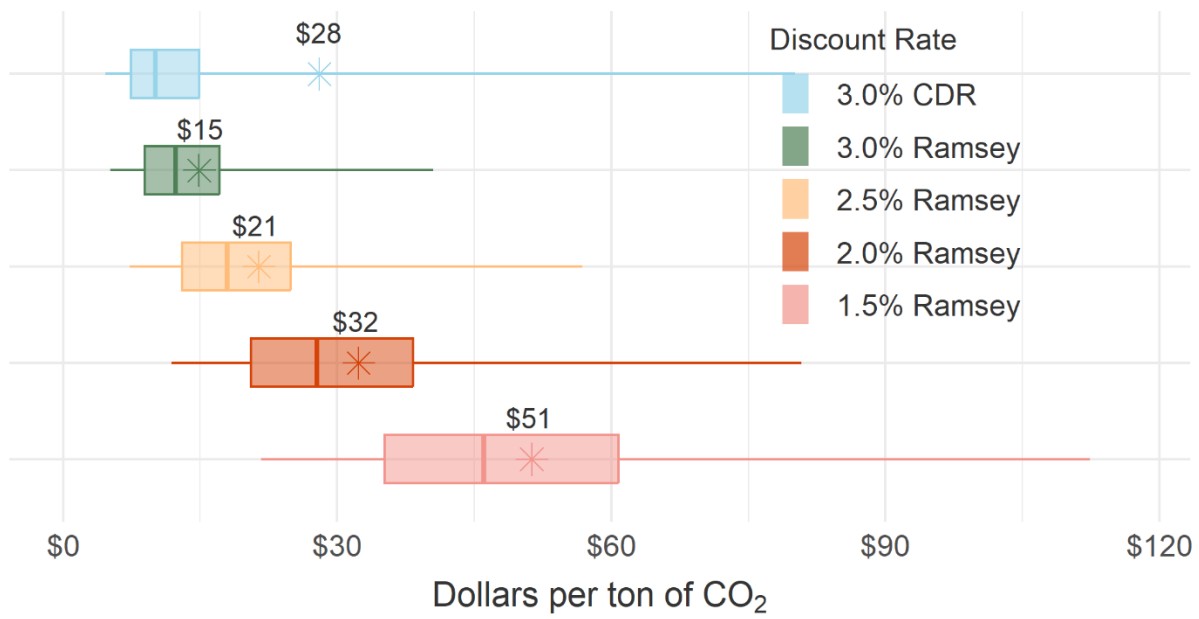

**Figure A54 2.** Net present value of future damages from one tonne of $CO_2$ for damages occuring only within the CONUS. Units are in dollars (2020 USD) per ton of $CO_2$ emitted. 'CDR' refers to Constant Discount Rate. Whiskers represent the 2.5th and 97.5th percentiles, while boxes span the 25th to 75th. Mean values (stars and text) along with median values (vertical lines) are also shown.


**Code Availability**

The Framework for Evaluating Damages and Impacts (FrEDI) is available on the U.S. EPA Enterprise GitHub https://github.com/USEPA/FrEDI/releases/tag/FrEDI_2300. FaIR is available at https://github.com/OMS-NetZero/FAIR. The RFF SP projections are available at https://zenodo.org/record/6016583 and the SSP projections

are available at https://tntcat.iiasa.ac.at/SspDb/.

**Data Availability**

All code and data associated with this study are available at www.github.com/USEPA/FrEDI_NPD.

**Author Contributions**

CH, EEM, MS drafted the manuscript text and figures with contributions from all co-authors. BP, EEM, KN, and CH

conducted the computational analysis. KN and JW developed the FrEDI code. SB drafted Figure 1 and provided

input on graphics and all authors contributed to the writing of this manuscript.

**Competing Interests**

The authors declare that they have no conflict of interest

**Acknowledgements**

The views presented in this manuscript are solely those of the authors and do not necessarily represent the views

or policies of the U.S. Environmental Protection Agency. Support for Industrial Economics was provided under EPA

contracts 47QFSA21D0002 and 140D0420A0002. The authors also wish to acknowledge research assistance and

other analytic support from William Maddock, Hayley Kunkle, Anthony Gardella, and Charles Fant.

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

Table 1.1.9. Implicit Price Deflators for Gross Domestic Product:
https://apps.bea.gov/iTable/?reqid=19&step=3&isuri=1&select_all_years=0&nipa_table_list=13&series=a&first_year=2006&last_year=2020&scale=-99&categories=survey&thetable=, last access: January 24, 2023.
U.S. Census: American Community Survey 2014-2018 [https://www.census.gov/programs-surveys/acs/data.html],
U.S. Environmental Protection Agency: Updates to The Demographic and Spatial Allocation Models to Produce Integrated Climate And Land Use Scenarios (ICLUS) (Version 2), Washington, DC, EPA/600/R-16/366F, 2017.