# Peer review of "Advancing the estimation of future climate impacts within the United States"

_EGUsphere, 2023_

## Referee Comment (RC2)

"Advancing the estimation of future climate impacts within the United States." *EGUSphere*, Manuscript #2023-114

**General comments**

This paper presents new estimates of the impacts of climate change on the economy and society of the United States. The authors assess both marginal and non-marginal changes in the climate, and present results that include a comprehensive list of sectors, ranging from human health to infrastructure and labor supply. Climate change impacts are disaggregated into seven regions across the country, with an additional analysis assessing the racial breakdown of total impacts. The framework they utilize – FrEDI – is built to adapt to an evolving scientific literature, such that damage functions can be updated or additional sectors added, as the evidence base improves.

This is an important line of inquiry, both for informing climate change mitigation and adaptation policy and for pushing climate economic research forward. While many climate impacts analyses exist, especially in the US, this appears to be the most comprehensive set of estimates that cover many sectors and are presented in a framework that allows for the calculation of total projected damages *and* marginal damages (i.e., a domestic social cost of carbon).

Unfortunately, the inputs to the FrEDI model, and some of its key features, are well behind best available science, making the findings difficult to trust. For example, adaptation to future climate change is largely assumed away (at least for all empirically derived damages), spatial resolution is exceptionally limited relative to other work, and key dimensions of uncertainty are ignored. While I think the fundamental goals and structure of FrEDI are valuable for research and policy, I do not think the results represent best available evidence on the question at hand.

I detail my specific comments below and provide some technical corrections and smaller questions at the end of my review.

**Specific comments**

1. Damage functions do not represent best available science

The damage functions that form the building block of FrEDI are outdated and, critically, fail to incorporate empirically-based estimates of adaptation. It is increasingly clear in a growing climate econometrics literature that populations adapt to a gradually changing climate (e.g., Auffhammer, 2018); generating projections that assume people will act in 2090 as if climate change hit them unexpectedly and without warning is unrealistic. This is particularly problematic in this study with respect to health, which completely dominates all projected damages. There is clear evidence that people adapt to temperature-driven mortality (Barreca et al., 2016; Heutel et al., 2021; Carleton et al., 2022), and yet this large literature is ignored and a damage function is used that assumes no adaptation (from Cromar et al., 2022). Based on prior work, damages via temperature-induced mortality are likely far too large in this manuscript due to this implausible assumption.

Other inadequacies with damage functions include:

- Damages are assumed to be proportional to income (as far as I can tell), which fails to account for the fact that future incomes are likely to dramatically lower sensitivity to climate extremes (e.g., Rode et al., 2021)
- Electricity demand is included but consumption of other energy sources, such as natural gas, are excluded. This, by construction, leads to an inflated projection of damages, as electricity is largely used for cooling while other energy sources are used for heating, demand for which will fall under a warming climate (Deschenes and Greenstone, 2011; Wenz et al., 2017; Rode et al., 2021)
- Uncertainty in damage functions appears to be ignored (e.g., see line 260), although it has been shown to play a critical role in overall uncertainty in prior work (Hsiang et al., 2017; Carleton et al., 2022)

2. Other features of FrEDI that fail to integrate best available science

There are two other features of FrEDI that fail to meet current literature standards.

- As far as I can tell, uncertainty in climate conditional on emissions is ignored (if this is not the case, it should be made much clearer how this is being handled). This uncertainty is large but also easily quantifiable using FaIR.
- Spatial heterogeneity in warming rates across the United States, as well as uncertainty in this spatial heterogeneity, are also ignored. This is unrealistic and easily remediable using available climate models.
- The spatial resolution of FrEDI, at just 7 regions across the U.S., fails to generate insights that can be used by local adaptation planners or policymakers, and fails to capture important local heterogeneities in exposure and vulnerability (which are particularly important in key sectors like health where damage functions are highly nonlinear in temperature).

3. Motivation is unclear

I am slightly confused by multiple claims made in the introduction regarding how this paper improves upon prior work. First, what exactly is meant by the "temperature binning approach" and why is it beneficial? If the authors intend to refer to the approach of reporting climate change impacts by warming levels (e.g., 2C by end of century), as opposed to reporting impacts by emission scenario (e.g., RCP4.5), this doesn't appear to be what is done throughout the paper. Moreover, such an approach doesn't "improve comparability between models" (line 59), it just hides this lack of comparability the background, as different scenarios and models will arrive at a given warming level under very different sets of assumptions.

Second, the authors claim that studies relying on the RCPs and SSPs are not run under "different future trajectories" (line 57), but this is not true. Most of these studies report impacts across the full ensemble of feasible RCPxSSP combinations; these are not probabilistic runs, but they are also not singular scenarios.

4. FrEDI faces key challenges as a "dynamic" framework

The authors describe FrEDI as a dynamic framework that can be updated over time as science evolves. However, this bottom-up framework that adds independently constructed sectors cumulatively to build estimates of a total impact of climate change will increasingly face two key challenges, both of which are left unaddressed by the authors. First, as more sectors are added, "double counting" of sectoral impacts becomes an increasing concern. This is likely already a problem in the current manuscript – labor hours lost and temperature-induced mortality likely overlap; labor supply and recreation likely overlap; flooding related traffic delays and damages associated with rail and roads likely overlap; etc. The authors should present a plan for addressing this issue both within this paper and in future applications of FrEDI, once more sectors are added.

Similarly, impacts in these sectors link to one another, but such interlinkages are ignored. For example, changes in the labor market will likely lead to population reallocations that shift health risks through demographically differentiated migration. Many other examples of interlinkages exist, and the importance of such links will only grow as more sectors are added to FrEDI. As with double counting, I think this issue should be addressed in this paper.

**Technical corrections/questions**

- Why avoid social cost of carbon (SCC) language when computing the net present value of a marginal ton? The authors are computing what the literature calls a "domestic SCC" – why not use this term to ensure consistency and clarity?
- Line 87 suggests socioeconomic and emissions scenarios are randomly and independently sampled, but my understanding of the RFF scenarios was that there is a joint distribution and that draws should therefore be jointly sampled.
- Line 243 – what figure is being referred to?
- It is very unclear how the racial breakdown was done – are these populations modeled as differentially vulnerable to the same physical hazards? Or are the authors simply calculating how these populations are distributed across the 7 regions? More detail on what these estimates do and do not include is needed.
- The Burke et al. (2015) citation on line 55 appears to be misplaced.
- Is it a feature or a bug that FrEDI can combine any socioeconomics with any warming scenario? Should these things be linked so as to ensure feasibility/plausibility? Lines 65-68)

**References**

Auffhammer, Maximilian. "Quantifying economic damages from climate change." *Journal of Economic Perspectives* 32, no. 4 (2018): 33-52.

Barreca, Alan, Karen Clay, Olivier Deschenes, Michael Greenstone, and Joseph S. Shapiro. "Adapting to climate change: The remarkable decline in the US temperature-mortality relationship over the twentieth century." *Journal of Political Economy* 124, no. 1 (2016): 105-159.

Carleton, Tamma, Amir Jina, Michael Delgado, Michael Greenstone, Trevor Houser, Solomon Hsiang, Andrew Hultgren et al. "Valuing the global mortality consequences of climate change accounting for adaptation costs and benefits." *The Quarterly Journal of Economics* 137, no. 4 (2022): 2037-2105.

Deschênes, Olivier, and Michael Greenstone. "Climate change, mortality, and adaptation: Evidence from annual fluctuations in weather in the US." *American Economic Journal: Applied Economics* 3, no. 4 (2011): 152-185.

Heutel, Garth, Nolan H. Miller, and David Molitor. "Adaptation and the mortality effects of temperature across US climate regions." *The review of economics and statistics* 103, no. 4 (2021): 740-753.

Rode, Ashwin, Tamma Carleton, Michael Delgado, Michael Greenstone, Trevor Houser, Solomon Hsiang, Andrew Hultgren et al. "Estimating a social cost of carbon for global energy consumption." *Nature* 598, no. 7880 (2021): 308-314.

Wenz, Leonie, Anders Levermann, and Maximilian Auffhammer. "North–south polarization of European electricity consumption under future warming." *Proceedings of the National Academy of Sciences* 114, no. 38 (2017): E7910-E7918.

---

## Community Comment (CC1)

[supplement omitted: unrelated document]

---

## Author Comment (AC2)

**Advancing the estimation of future climate impacts within the United States**

**Response to Reviewers**

**Reviewer #1**

The manuscript describes a framework for evaluating long-term damages from climate change to a large number of different specific U.S. economic sectors. The damage functions have generally been previously published and have been extended to 2300 (from an original 21st century time horizon for most of them). The common FaIR climate model is used to simulate and take into account uncertainty in global climate, specifically temperature change, which is used as input to the various damage functions. Probabilistic projections for socioeconomic conditions (e.g., GDP, population change) from a recent study are used, and combined with different emissions scenarios. The work decomposes climate-driven damages by geographic region within the U.S. and by different vulnerable populations.

The manuscript is well-written and well-organized. As near as I can tell from the methodological descriptions of FrEDI provided in the manuscript, the conclusions and results follow nicely from the experiments that are conducted. However, my comments below revolve mostly around the theme of a need for further details about the structural assumptions baked into the damage function underlying FrEDI. You understandably defer a great deal to other recently published works that are the primary sources for these parameterizations, and I understand that this is not a model description paper. However, this makes it a bit more difficult for the reader to understand the context for the results presented, particularly for interpreting sector-specific damages. Most of these and my other comments I anticipate can be cleared up through adding some modeling details and/or caveats. Thus I recommend for minor revisions.

*We thank Reviewer #1 for their comments, particularly related to our discussion regarding the treatment of adaptation and differential impacts in FrEDI. These have helped us improve the quality of our methods and discussion of results. We have addressed each comment in detail below in italics.*

 **General comments:**

L65-67 - Perhaps any of these FAIR warming scenarios *can* be paired with any socioeconomic projection, but *should* they? There are some SSP-RCP combinations (e.g.) that modelers use but shouldn't. For example, see O'Neill et al 2020, https://doi.org/10.1038/s41558-020-00952-0, particularly their Figure 2. I can imagine that your probabilistic framework may account for this in some way, but I'd be interested to know more about how you ensuring that unlikely/implausible combinations of scenarios aren't unduly biasing results?

*Thank you for the comment. Both reviewers had similar comments with how the current text is written with respect to how we pair RFF-SPs scenarios with FaIR. We will update the text with a more complete description. Here is a description of the process conducted that will be included in the manuscript: Uncertainty from climate is treated independently from uncertainty in the emissions and socioeconomics. Within this study we are passing projected emissions of $CO_2$, $N_2O$, and $CH_4$ from the RFF-SPs to the FaIR model to calculate a temperature trajectory from each emission pathway. The FaIR model developers have provided 2,237 sets of climate parameters, calibrated to the IPPC AR6. In this work, we use the*

*Monte Carlo simulation capabilities of MimiGIVE.jl (https://github.com/rffscghg/MimiGIVE.jl) to randomly sample the RFF-SPs and FaIR parameter sets to generate 10,000 global temperature trajectories. The resulting temperature trajectory associated with each of the RFF emissions pathways are then paired with the corresponding RFF socioeconomic (U.S. GDP and population) trajectories, which are then passed into FrEDI to calculate the physical and economic impacts from each RFF-SP scenario.*

L74, 159 - What assumptions about coastal adaptation (and other forms of adaptation/mitigation, and development) are baked into FrEDI? Can you comment on how realistic these are on such long time scales?

*FrEDI maintains adaptation assumptions from the underlying studies that form the basis of FrEDI's temperature-driven sectoral damage functions. For most of these studies, because the implicit or explicit impact response function is calibrated to historical or current data, this means that historically practiced adaptation or hazard avoidance actions are "baked in" – but enhanced adaptation action, or new (currently unknown) technologies are not considered. The exceptions include coastal property and select other infrastructure sectors, where the underlying studies consider specific adaptation actions. These have been incorporated into FrEDI.  For example, for the coastal flooding sector, FrEDI's default adaptation assumption is a Reactive Adaptation scenario, as defined in Neumann et al. (2021), and includes the costs (and reflects the hazard reduction benefits) of elevation of properties, where and when the benefits exceed the costs of this measure and expanded beach nourishment at locations where it is currently practiced. No other measures are included. There is an option in FrEDI, however, for the user to select either a No Adaptation scenario for this sector, which excludes the option to elevate properties as well as measures that might hold back floodwaters, or a Proactive Adaptation scenario, where adaptation measures include elevation, beach nourishment, and armoring (either with bulkheads in protected areas or more expensive seawalls in areas exposed to higher open ocean wave action). It is difficult to comment on the realism of future action. There is some discussion in both Neumann et al. (2021) and Lorie et al. (2020), both of which make the point that even under current coastal hazards, cost-effective adaptation measures have not been adopted, probably because they involve short term capital investment in order to yield future, uncertain benefits. This is one reason why Proactive adaptation is not the default scenario in FrEDI. In Table A3, we present the climate-driven damages under different adaptation assumptions, where available, to explore the sensitivity to adaptation and highlight this capability within FrEDI.*

L186 - Related to above, looking specifically at marginal damages means that the assumed adaptation strategies could be masking the real magnitude of these climate risks. For example, if it is assumed that you make least-cost decisions, then these marginal damages will be more of a lower bound. While the adaptation assumptions are provided in an appendix, it may be worth discussing these in main text more, because (my impression is that) the results rely a great deal on these assumptions.

*In this analysis, we estimate the marginal damages by holding the adaptation scenario (that is, No Adaptation, Reactive, or Proactive) constant overtime, for those sectors where an option is provided. For other sectors, historically practiced adaptation or hazard avoidance actions are "baked-in" as described above. Therefore, while it is certainly true that marginal damages are higher under 'No Adaptation' versus 'Reactive' or 'Proactive' Adaptation scenarios, climate-driven damages under FrEDI's default adaptation assumptions are likely not masking the magnitude of climate risks, given the assumption that current mitigation levels will be held constant in the future. However, the user has a choice to explore*

*different adaptation options in select sectors. We will update the main and supplemental text to clarify the default adaptation assumptions included in this analysis.*

*Note also that the sea level rise module uses financial smoothing so that costs are spread out over time to avoid discontinuities that could lead to odd effects with marginal damage calculations.*

*Related to the comment on coastal adaptation, the authors have separately done research investigating how to implement sub-optimal adaptation choices that might better represent real world behavior (see Lorie et al, 2020, https://www.ncbi.nlm.nih.gov/pmc/articles/PMC7433032/). FrEDI's capability to analyze three different assumptions about adaptation for many sectors allows exploration of more and less optimistic assumptions. It is difficult to determine if overall, the default FrEDI assumptions are too optimistic about rational adaptation, or include insufficient adaptation possibilities.*

L210-212 - It isn't totally clear here what the default adaptation assumptions are for each sector (maybe cite Table A2 here?), or what each of these adaptation assumptions represents with respect to the individual sectors. For example, what does "reasonably anticipated adaptation" mean for coastal properties? And how reasonable are these assumptions on such long time scales, for each sector?

*Thank you for the comment. We added a reference to Neuman et al., 2021 in table A2 and expanded on the meaning of these adaptation options within the text. Reasonably Anticipated Adaptation is an option for the High-Tide Flooding/Coastal Roads sector, but not for coastal properties. This adaptation scenario models reduced road delays associated with drivers seeking alternative, non-flooded routes (which still results in delays compared with conditions where flooding is not present, but delays of a much lower magnitude than under No Adaptation); it also includes the benefits of ancillary protection when/if nearby seaward properties would be likely to implement protective measures to hold back flooding (because benefits of armoring measures exceed costs at those properties). Reasonably Anticipated Adaptation is an "intermediate" adaptation assumption – it yields much lower costs than the No Adaptation option and higher costs than the Proactive Adaptation option.*

**Specific comments:**

L23-27 - Is this by the year 2090? Or when?

*Yes, the annual climate driven damages are for 2090. The text will be updated to reflect this.*

L50 and 52 - "Climate Change Impacts and Risk Analysis (CIRA) project" and "Climate Impact Lab" - I'm aware of these projects/groups, but I'm not sure readers in general will know the context for these. Could be worth specifying where they're housed.

*Thank you for the suggestion. The following text will be included in the manuscript: For example, the Climate Change Impacts and Risk Analysis (CIRA) project, coordinated by the USEPA involving researchers from government, academics and consultants, quantifies the physical effects and economic damages of climate change in the U.S., using detailed models of sectoral impacts (e.g., human health, infrastructure, and water resources) (EPA, 2017a). In addition, the Climate Impact Lab is a collaboration of more than 30 climate scientists, economists and researchers from across the U.S. and has focused its work on understanding the economic damages from climate change within the U.S. (Hsiang et al., 2017) and*

*across the globe, including impacts to human health (Carleton et al., 2022), agriculture (Rising and Devineni, 2020; Hultgren et al., 2022), coastal property (Depsky et al., 2022), and energy (Rode et al., 2021).*

L70 - In what way(s) is FrEDI distinguishable from a traditional IAM?

*FrEDI is similar to traditional IAMs in that it tries to link society and the economy with climate change into one modeling framework. However traditional IAMs include economic processes as well as processes that produce greenhouse gases. Simple IAMs compare costs and benefits of avoiding levels of warming. Other, more complex IAMs, investigate processes that cause or prevent greenhouse gas emissions, such as, energy technologies, energy use choices, and land-use changes. These models will typically trade off to find optimal policies to reduce emissions. Within this analysis, emissions are an input to FaIR, which produces temperature change for an input to FrEDI.*

L114-116 - Are there or should there be correlations and/or feedbacks represented between the RFF-SP emission scenarios and the FaIR scenarios?

*Following up on the response in the first comment, no there are no correlations and/or feedbacks between the RFF-SPs and the FaIR uncertainty. There are two types of uncertainty explored here, the socioeconomic and emission uncertainty coming from the RFF-SPs and the physical climate parameter uncertainty coming from FaIR. These are treated independently from each other by using the Monte Carlo simulation capabilities of MimiGIVE.jl (https://github.com/rffscghg/MimiGIVE.jl) to randomly sample the RFF-SP emission and FaIR parameter sets. There is the possibility that high climate sensitivities could lead to implementation of climate policies (Webster et al. 2008), however, we do not include this in our analysis. The analysis presented within this paper is a common method for assessing future uncertainty and can be found throughout WG3 of the IPCC, within Rennert et al., 2022 using the RFF-SPs and the FaIR climate model, Yang et al., 2021 using GCAM and the climate model MAGICC to assess probability of staying below 2°C. We have clarified this in the main text.*

*Rennert et l., 2022, "Comprehensive evidence implies a higher social cost of $CO_2$". Nature.*

*Yang et al., 2021, "Can updated climate pledges limit warming well below 2°C?". Science.*

L116-118 - How sensitive are the results to this moderate assumption? And no uncertainty sampled in these other gases/aerosols.

*There has been some recent work to be able to infill scenarios with gases and aerosols by matching to the WG3 databases of 1000s of emission scenarios to $CO_2$, $CH_4$ and $N_2O$ emissions (https://silicone.readthedocs.io/en/latest/# ). We did not include this within our current analysis and is a potential source of uncertainty. We have a separate project working on using the Silicone model (Lamboll et al. 2020) to extrapolate to other gases and aerosols. The early indication of this work is that the effects are not large: generally, aerosols are projected to decrease across almost all scenarios, so there isn't much radiative forcing spread between the high and low aerosol scenarios (and the other GHGs are small contributors relative to big three).*

L175 - should this be g_t?

*Yes this was a typo and will be updated.*

Table 2 - should "2,1" be "2,100" in the 95% CI column?

*Yes this was a typo and will be updated.*

L270-272 - I'm intrigued by this and would love to see more work on this prominently featured in contemporary research. But I'm also curious about how this modeling is done within FrEDI and what scale data is being used, and what assumptions there are when disaggregating damages for socially vulnerable populations.

*Thank you for the comment. We will move much of the current social vulnerability analysis from the Supplement to a new section in the main text and provide more background and detail about this capability. We provide a general overview here.*

*Differential climate change risks are a function of exposure to where climate change impacts are projected to occur and vulnerability, in terms of an individual's capacity to prepare for, cope with, and recover from climate change impacts ( https://www.epa.gov/cira/social-vulnerability-report). The capability of exploring differential impacts to U.S. populations within FrEDI  is based on data on the locations of where populations live as an indicator of exposure and for vulnerability, considers four categories for which there is evidence of differential vulnerability. Differential impacts in each group are calculated in FrEDI at the Census tract level as a function of current population demographic patterns (i.e., percent of each group living in each census tract), projections of CONUS population (from ILCUS, U.S. Environmental Protection Agency, 2017), and projections of where climate-driven impacts are projected to occur (i.e., using FrEDI temperature-impact relationships) at the Census tract level.  We assess impacts in FrEDI across four different population categories: income, age, race and ethnicity, and education, and breaking the race and ethnicity category into BIPOC populations. Results show that the BIPOC group is more likely to be impacted by climate-driven changes to air quality attributable mortality as well as lost labor hours due to increasing temperatures. Breaking down the results further, Black or African Americans are more likely to experience additional pre-mature mortality from climate-driven changes in air quality, and Hispanic or Latino are more likely to experience additional lost labor hours.*

L424-430 - You cite two studies that used a power law relationship, finding that it fit the temperature-damage relationship better than linear. Why choose to use linear instead in FrEDI?

*Sarofim et al., 2021, showed that for most of the analyzed sectors, a linear fit best described the relationships between temperature and damages were linear.  However, we are using a piecewise linear function for FrEDI, where a damage function is calculated within every temperature bin (1-2°, 2-3°, etc.), FrEDI can capture nonlinearities resulting from the underlying impact models up to the largest temperature changes assessed in the underlying studies. As described in the main text and Section A1, FrEDI's damages functions are extended to warmer temperatures (i.e., later years), by linearly extrapolating each sectoral damage function.*

Figure A2 - There is a lot to unpack here, including a few of what appear to be some tipping points or other kinds of jumps in some of these sectoral damages around 2060 or so. E.g., wind damage, agriculture, or the sudden jump in coastal property damages after about 2080 - how do you make sense of these results, in light of the adaptation assumptions embedded in FrEDI?

*We provide Figure A2 as an illustration of the overall trends and relative magnitudes of damages projected to occur within each impact sector currently included in FrEDI. As described in the FrEDI technical documentation (https://www.epa.gov/cira/fredi), the damages in many of these sectors are a function of the climate driver (e.g., temperature or sea level change), as well as additional scaling factors, such as socioeconomics  (e.g., damages resulting from mortality rely on the Value of a Statistical Life, which is proportional to per capita income). The temporal trends in Figure A2 reflect the combination of the projected changes in both the climate drivers and these scalars, the latter of which exhibit relatively smooth trends (Figure A1). In contrast, damages in other sectors are only a function of climate drivers (e.g., agriculture, wind damage, coastal properties) and therefore the trends in Figure A2 are more directly reflective of the underlying by-degree piece-wise linear temperature (or sea level rise) binned damage functions. The slight discontinuities pointed out in some of these sectors (e.g., agriculture) can occur either at the boundary between temperature bins (e.g., for agriculture and wind damage) or due to thresholds in the underlying studies. For example, the sharp increase in damages in the coastal property damage sector after 2080 are directly reflected in the underlying damage functions (See FrEDI Technical Documental Appendix B) and correspond to a sharp increase in damages that occur after sea levels breach 100 cm. These trends also reflect the default adaptation assumptions included in FrEDI. For example, the damage functions for the default reactive adaptation scenario for the coastal property sector include large increases in damages above 100 cm of sea level rise, whereas the proactive adaptation scenario, which includes collective action such as seawall construction, does not include the same rate of damage increase at 100cm of sea level rise. Details about the underlying studies and damage function development details for each sector are included in the FrEDI Technical Documentation.*

*We have added a brief explanation in the caption of Figure A2.*

L469 - perfect foresight over what time frame?

*Yes, thank you: different sectors have different time frames: e.g., the coastal property model does a 30 year look ahead for infrastructure spending decisions. We've edited the sentence to remove the phrase "perfect foresight".*

Figure A5 - Are some of these damage distributions in later years so heavily skewed that the means are outside of the 95% CI? It might be tidier to display the median, and perhaps more consistent with the use of percentiles for the 95% CI too.

*We present two figures within the Appendix that have been updated with mean and median lines. Citing Webster et al., 2008, under a risk based framework, the most important causes for concern are not the median projections of future climate change, but the low-probability, high-consequence impacts. Therefore, we present the mean damages over the median, as well as the 95% confidence rank.  We note that in figure A5, the mean is much larger than the median due to significant growth projected to occur in the socioeconomics (GDP/capita) between2100 and 2300. In these cases, a small number of high impact scenarios are resulting in the mean damages exceeding the 97.5th percentile. Below are the new versions of figures that will be included in the Appendix.*

Annual Climate-Driven Impacts

[Figure]

Annual Climate-Driven Impacts

---

## Author Comment (AC3)

**Advancing the estimation of future climate impacts within the United States**

**Response to Reviewers**

**Reviewer #2**

"Advancing the estimation of future climate impacts within the United States." EGUSphere, Manuscript #2023-114

General comments This paper presents new estimates of the impacts of climate change on the economy and society of the United States. The authors assess both marginal and non-marginal changes in the climate, and present results that include a comprehensive list of sectors, ranging from human health to infrastructure and labor supply. Climate change impacts are disaggregated into seven regions across the country, with an additional analysis assessing the racial breakdown of total impacts. The framework they utilize – FrEDI – is built to adapt to an evolving scientific literature, such that damage functions can be updated or additional sectors added, as the evidence base improves.

This is an important line of inquiry, both for informing climate change mitigation and adaptation policy and for pushing climate economic research forward. While many climate impacts analyses exist, especially in the US, this appears to be the most comprehensive set of estimates that cover many sectors and are presented in a framework that allows for the calculation of total projected damages and marginal damages (i.e., a domestic social cost of carbon).

Unfortunately, the inputs to the FrEDI model, and some of its key features, are well behind best available science, making the findings difficult to trust. For example, adaptation to future climate change is largely assumed away (at least for all empirically derived damages), spatial resolution is exceptionally limited relative to other work, and key dimensions of uncertainty are ignored. While I think the fundamental goals and structure of FrEDI are valuable for research and policy, I do not think the results represent best available evidence on the question at hand.

I detail my specific comments below and provide some technical corrections and smaller questions at the end of my review.

We thank Reviewer #2 for providing constructive comments and positive feedback that the fundamental goals and structure of FrEDI are valuable for research and policy. These have greatly helped us to refine and improve the clarity of our key points and methods, as well as improve our discussion and analysis related to the treatment of adaptation options and uncertainty characterization within FrEDI . We will walk through our responses to the major concerns below and include additional figures within the comments.

1.  "well behind best available science" *Many of the underlying studies used to develop damage functions for FrEDI have been published since 2017. The most recent study to be incorporated into FrEDI is Cromar et al., 2022, which calculates temperature-related mortality and is also used within the GIVE model (Rennert et al. 2022). We strive to bring in and update FrEDI with the latest and best available science that currently exists and can be adapted to FrEDI temperature (and sea level rise) binning approach.*

2. "adaptation to future climate change is largely assumed away" *There are three categories of adaption options for select sectors within FrEDI, which are based on the available information in the underlying impact studies. First, there are five sectors (i.e., coastal property, roads, etc.) that explicitly explore alternative adaptation options (i.e., proactive and reactive). These options are available within FrEDI and are explored in more depth in section A4. Secondly, there are studies that do not explicitly include different adaptation scenarios but do include adaptation measures within the sectoral model. For example, in the electricity supply and demand sectoral analysis from McFarland et al., 2015 has air conditioner penetration within the sectoral models. Lastly, there are some sectors (i.e., marine fisheries) that do not explore additional adaptation options. Damages in this last category largely reflect the mitigation levels present during the time period of the analysis.*

3. "spatial resolution is exceptionally limited" *The spatial resolution for FrEDI v3.0 contains 7 regions within the CONUS. Each sector's damage functions are aggregated from highly spatially resolved sectoral modeling. Many of the sectors use the LOCA downscaling dataset for the climate model data ($1/16^{th}$ degree resolution), and then the impact models are run at county, census block, or other resolution. For example, the Neumann et al.,2021 study explores roads, rail, and coastal property at the county resolution, while the Fann et al., 2021 study explores air quality damages at 36km resolution. In order to maintain a flexible, modular, and computationally efficient model, FrEDI uses a relationship between temperature and damages at the NCA region level rather than the native resolution of the underlying impact model, allowing us to be able to conduct studies such as this one where uncertainties across tens of thousands of scenarios can be explored. Similar models in this space, like DSCIM and GIVE also trade spatial resolution for computational efficiency.*

4. "key dimensions of uncertainty are ignored" *Thank you for bringing up uncertainty. We will expand our text to include a more thorough analysis of the uncertainty characterization that we can explore within FrEDI. In addition to climate parameter, emission, and economic uncertainties already explored in the analysis, FrEDI also includes the structural uncertainty in damage functions within and across the three temperature-related mortality studies included in FrEDI. This sector is the largest single impact category in our analysis. FrEDI already includes three distinct damage function estimates from three distinct studies and includes high and low confidence intervals for two of these (based on the information in each underlying study). The physical impacts from these three estimates in 2090 can differ by over a factor of 10 and can be sensitive to available adaptation assumption. We chose to use the Cromar et al., damage function as our default temperature-related mortality function in order to align with the GIVE model. While this function yields significantly larger results than the other two functions, the resulting damages in 2090 are not inconsistent with similar previous published studies (see below response). Within the FrEDI framework, we are continually working towards including more sectoral studies from different authors to be able to further characterize these types of uncertainties. We have added additional discussion to the main text and supplement presenting these results to help to better characterize this additional aspect of uncertainty.*

Specific comments

1. Damage functions do not represent best available science

   The damage functions that form the building block of FrEDI are outdated and, critically, fail to incorporate empirically-based estimates of adaptation. It is increasingly clear in a growing climate econometrics literature that populations adapt to a gradually changing climate (e.g., Auffhammer, 2018); generating projections that assume people will act in 2090 as if climate change hit them unexpectedly and without warning is unrealistic. This is particularly problematic in this study with respect to health, which completely dominates all projected damages. There is clear evidence that people adapt to temperature-driven mortality (Barreca et al., 2016; Heutel et al., 2021; Carleton et al., 2022), and yet this large literature is ignored and a damage function is used that assumes no adaptation (from Cromar et al., 2022). Based on prior work, damages via temperature-induced mortality are likely far too large in this manuscript due to this implausible assumption.

   *See responses above re 'the best available science' and adaptation options.*

   *We agree with the reviewer that the lack of additional adaptation within the Cromar et al., 2022 is not ideal. However, when exploring the physical impacts from multiple temperature-related mortality studies we find that by the end of the century, the estimates from FrEDI for premature mortality are in line with several other studies. For example, FrEDI calculates 19,000 to 91,000 premature deaths with a mean of 50,000 in 2090 from the Cromar et al., 2021 study. Shindell et al., 2020 (https://www.ncbi.nlm.nih.gov/pmc/articles/PMC7125937/) finds 100,000 deaths in 2100, even after including adaptation options. The Lancet Health Explorer calculates 84,000 deaths in 2100 from SSP3-7.0, which builds upon the 2022 report (Romanello et al., 2022). (https://climatevulnerabilitymonitor.org/health/usa/heat-related-impacts/heat-related-mortality/ ).We are, however, communicating with the authors of Carleton et al. to investigate whether the underlying data in their analysis is compatible with the FrEDI damage function approach, and given that the temperature-related mortality sector is so important, are continuing to pursue additional options. I don't know that either is the gold standard of studies, but they show that Cromar et al. is not alone at that high end, and they are all reputable research groups.*

   Other inadequacies with damage functions include:
   • Damages are assumed to be proportional to income (as far as I can tell), which fails to account for the fact that future incomes are likely to dramatically lower sensitivity to climate extremes (e.g., Rode et al., 2021)

   *While the health sectors that use a value of statistical life are based on GDP (i.e., air quality, temperature related mortality, wildfire, valley fever, southwest dust), not all sections within FrEDI are a function of GDP. We will clarify in the text which sectors are proportional to GDP (Table A1). While there may be evidence that wealthier societies may be less sensitive to climate extreme, we also note evidence that  the number of billion-dollar disasters from NOAA are*

*trending upward even though the US is getting wealthier.*
*https://www.ncei.noaa.gov/access/billions/time-series*

*We can also note that there is some evidence that historical trends in some kinds of infrastructure-based damages appear to increase roughly proportionally to GDP – see, e.g., research on normalizing hurricane damage by GDP (Grinsted et al., 2019, https://www.pnas.org/doi/10.1073/pnas.1912277116). While in theory improvements in technology, building codes, weather forecasting, and other aspects of societal advancements would be expected to reduce damages relative to GDP, there appear to be other factors involved that seem to counteract such benefits.*

• Electricity demand is included but consumption of other energy sources, such as natural gas, are excluded. This, by construction, leads to an inflated projection of damages, as electricity is largely used for cooling while other energy sources are used for heating, demand for which will fall under a warming climate (Deschenes and Greenstone, 2011; Wenz et al., 2017; Rode et al., 2021)

*The damage function for electricity demand and supply in FrEDI v3.0 is developed from McFarland et al., 2015 (Climatic Change).  The study accounted for changes in both heating and cooling degree days, accounting for different fuel sources. One of the models used in McFarland et al., 2015 is the GCAM model modeling four types of fuels including electricity, natural gas, fuel oil and biomass.*

• Uncertainty in damage functions appears to be ignored (e.g., see line 260), although it has been shown to play a critical role in overall uncertainty in prior work (Hsiang et al., 2017; Carleton et al., 2022)

*Uncertainty in damage functions is typically not well represented in of the underlying sectoral modeling studies used to develop FrEDI damage functions.  However, FrEDI has included confidence interval ranges for temperature-related mortality that were available from the from Cromar et al., and Hsaing et al., studies. The confidence intervals between these studies overlap, but estimate different extremes. For instance, the Cromar-derived damage function estimates net damages from temperature-related mortality, while the low confidence range for the Hsiang-derived damage function has a small potential for net benefits in this sector in 2090.*

*In addition to uncertainties within each damage function, FrEDI also has the capability to explore some structural uncertainty by providing incorporating multiple studies for the same sector (i.e., temperature-related mortality).  We will include a section in the Appendix that evaluates total mean climate damages in 2090 under the three temperature-related mortality estimates, as well as show the confidence intervals for the 2 studies in 2090.*

| 2090 Premature mortality – Billions  USD | | | |
|---|---|---|---|
| Model | 5th | 95th | Mean |
| Cromar et al., | 300 | 3,900 | 2,100 |
| Hsiang et al., | -280 | 1,800 | 740 |

| Mills et al., (w/ adaptation) | - | - | 31.0 |
|---|---|---|---|
| Mills et al., (w/o adaptation) | - | - | 110 |

2. Other features of FrEDI that fail to integrate best available science There are two other features of FrEDI that fail to meet current literature standards.

• As far as I can tell, uncertainty in climate conditional on emissions is ignored (if this is not the case, it should be made much clearer how this is being handled). This uncertainty is large but also easily quantifiable using FaIR.

*We will update the text to provide more detail on how the analysis was conducted.* Please see our response to a similar question raised by Reviewer #1.

• Spatial heterogeneity in warming rates across the United States, as well as uncertainty in this spatial heterogeneity, are also ignored. This is unrealistic and easily remediable using available climate models.

*FrEDI embodies spatial heterogeneity within the underlying damage functions. Uncertainty in these spatial patterns is partially captured by the different GCMs from the underlying studies. In this analysis, we use the average of the temperature/damage relationship derived from running the output of the different GCMs through the various impact models. Sarofim et al., 2021 finds that for the sectors that contain estimates for multiple GCMs (e.g., labor, electricity demand and supply, southwest dust), the damage functions are similar across GCMs for each degree of warming. Therefore, we use the average of the impact models within this analysis.*

• The spatial resolution of FrEDI, at just 7 regions across the U.S., fails to generate insights that can be used by local adaptation planners or policymakers, and fails to capture important local heterogeneities in exposure and vulnerability (which are particularly important in key sectors like health where damage functions are highly nonlinear in temperature).

*FrEDI was not designed with local adaptation planners in mind, and instead developed to rapidly produced national and regional scale estimates under different temperature pathways. We appreciate the comment that as FrEDI is now with only 7 regions it may not be applicable to local adaptation planners or policymakers. We will keep this in mind as we develop FrEDI in the future.*

*The spatial resolution for FrEDI v3.0 contains 7 regions within the CONUS. Each sector's damage functions are developed from highly spatially resolved sectoral modeling (see above response referencing LOCA). For example, the Neumann et al.,2021 study explores roads, rail, and coastal*

*property at the county resolution. In order to maintain a flexible, modular, and computationally efficient model, we have aggregated the detailed spatial variability data in the underlying studies in order to build a flexible and computationally inexpensive framework. This allows us to be able to conduct studies such as this one where 1000s of scenarios can be simulated. Similar models in this space, like DSCIM and GIVE also trade in spatial resolution for computational efficiency.*

3. Motivation is unclear

I am slightly confused by multiple claims made in the introduction regarding how this paper improves upon prior work. First, what exactly is meant by the "temperature binning approach" and why is it beneficial? If the authors intend to refer to the approach of reporting climate change impacts by warming levels (e.g., 2C by end of century), as opposed to reporting impacts by emission scenario (e.g., RCP4.5), this doesn't appear to be what is done throughout the paper. Moreover, such an approach doesn't "improve comparability between models" (line 59), it just hides this lack of comparability the background, as different scenarios and models will arrive at a given warming level under very different sets of assumptions.

*Thank you bringing this to our attention. After re-reading the manuscript we have deleted the text about temperature binning. The temperature binning approached used within FrEDI is captured within Sarofim et al., 2021. Essentially, impacts are binned into degree integer bins. A damage function is calculated for each degree bin and then interpolated between bins. This methodology allows for more comparability between studies and not bounded by the particular scenario within the underlying study. We have also updated our text throughout the introduction and abstract to clarify and refine our key points and the novel aspects of this approach.*

Second, the authors claim that studies relying on the RCPs and SSPs are not run under "different future trajectories" (line 57), but this is not true. Most of these studies report impacts across the full ensemble of feasible RCPxSSP combinations; these are not probabilistic runs, but they are also not singular scenarios.

*The underlying studies used within FrEDI typically use two RCPs trajectories and a few climate models. For example, Neuman et al., 2021 uses RCP4.5 and RCP8.5 across 5 climate models to assess the climate effects on costal properties, roads, and rail. These studies can't be used to assess marginal changes, policy pathways, or scenarios such as the Nationally Determined Contributions. After reading through the introduction again, we decided that this information was not needed within the text and have deleted it. We thank the reviewer for bringing this to our attention.*

4. FrEDI faces key challenges as a "dynamic" framework
   The authors describe FrEDI as a dynamic framework that can be updated over time as science evolves. However, this bottom-up framework that adds independently constructed sectors

cumulatively to build estimates of a total impact of climate change will increasingly face two key challenges, both of which are left unaddressed by the authors. First, as more sectors are added, "double counting" of sectoral impacts becomes an increasing concern. This is likely already a problem in the current manuscript – labor hours lost and temperature-induced mortality likely overlap; labor supply and recreation likely overlap; flooding related traffic delays and damages associated with rail and roads likely overlap; etc. The authors should present a plan for addressing this issue both within this paper and in future applications of FrEDI, once more sectors are added.

*Thank you for the comment. We agree that double counting of impacts poses a real problem and are very aware of this problem as we add in additional sectors. For example, FrEDI contains 3 different temperature-related mortality studies, and only one is used to aggregate up to regional and national impacts. There are also 2 roads studies, one that includes different kinds of road material and the other one that only explores impacts to asphalt roads. We take care to not include both road estimates when we are aggregating up to regional or national totals and include data flags and documentation to help FrEDI users avoid double counting as well. While the examples provided above may have overlapping effects, the sectoral studies do not include these overlapping effects. As a dynamic framework we can refine existing damage functions and add new damage functions to reflect the latest available science. Within the section A.2 of the technical documentation, we lay out a table of how we incorporate new studies and sectors within FrEDI. www.epa.gov/cira/fredi*

Similarly, impacts in these sectors link to one another, but such interlinkages are ignored. For example, changes in the labor market will likely lead to population reallocations that shift health risks through demographically differentiated migration. Many other examples of interlinkages exist, and the importance of such links will only grow as more sectors are added to FrEDI. As with double counting, I think this issue should be addressed in this paper.

*We agree with the reviewer here that we are missing some potentially large interlinkages and interdependencies. The research community as a whole is pushing further into this space; however it still remains a challenge to model. As studies that explore these linkages become available, we will look for ways to add them to FrEDI. We will add in more language caveating this to the manuscript.*

*We disagree with specific double counting issues raised by the reviewer: labor hours lost is based solely on living individuals working fewer hours, and therefore should not have much overlap with temperature-induced mortality (in the unfortunate cases where outdoor laborers pass away, they would be replaced by new laborers who would have similar responses to elevated temperatures). Any connection between reduced labor hours and the various recreation sectors in FrEDI will be tenuous: perhaps laborers who reduce working hours due to high temperatures might want to avail themselves of outdoor recreation, but this would not be anticipated to have a substantial effect relative to the total impacts on recreation and labor by themselves, and if anything, would lead to increased damages due to more demand for the degraded recreational supply. Finally, the road infrastructure analysis only considered temperature and precipitation-*

*based impacts on roads, and so would be mostly orthogonal to the coastal sea level surge-based flooding traffic delay analysis. We agree that in the future, there will be more potential for double-counting, and will need to address those possibilities carefully.*

Technical corrections/questions

- Why avoid social cost of carbon (SCC) language when computing the net present value of a marginal ton? The authors are computing what the literature calls a "domestic SCC" – why not use this term to ensure consistency and clarity?

*We use the term net present damage to avoid any potential confusion with the governmental SC-GHG process, though "net present damages of an additional ton" is indeed a synonym with the term "domestic SCC". Because we think that the global SC-GHG is the appropriate value to be used for cost-benefit, we wanted to avoid any implication that the domestic net present damages shown here should be considered a potential substitute in that kind of analysis. We will add a footnote clarifying that these numbers are comparable.*

- Line 87 suggests socioeconomic and emissions scenarios are randomly and independently sampled, but my understanding of the RFF scenarios was that there is a joint distribution and that draws should therefore be jointly sampled.

*We will clarify the dependance between the socioeconomic and emissions pathways (they are coupled draws). "we first utilize 10,000 paired probabilistic emissions and socioeconomic projections and a reduced-complexity climate model to provide inputs for FrEDI to assess the annual physical and economic impacts of climate change projected to occur through the end of the 21$^{st}$ century within 20 sectors in the contiguous United States (CONUS)."*

*Only the climate uncertainty is independently sampled.*

- Line 243 – what figure is being referred to?
*We will update the text to reflect the correct figure.*

- It is very unclear how the racial breakdown was done – are these populations modeled as differentially vulnerable to the same physical hazards? Or are the authors simply calculating how these populations are distributed across the 7 regions? More detail on what these estimates do and do not include is needed.
*Both reviewers commented on needing more information on social vulnerability analysis. We will add a new section to the paper that lays out the details and presents the figure in the main text and not in the Appendix. Please see our response to a similar request from Reviewer #1.*

- The Burke et al. (2015) citation on line 55 appears to be misplaced.
*We deleted this reference.*

- Is it a feature or a bug that FrEDI can combine any socioeconomics with any warming scenario? Should these things be linked so as to ensure feasibility/plausibility? Lines 65- 68)

*Thank you for bringing this to our attention. We do intend for the GDP and population scenarios that are inputs to FrEDI to be consistent with the emission scenario that is used to create the temperature projections that are also inputs to FrEDI.  In this study the projected temperatures resulting from running a given emission scenario through the FaIR model are matched with the GDP and population corresponding to that emission scenario (based on the RFF scenarios).  We could include some text within FrEDI's technical documentation cautioning the user.*

References

Auffhammer, Maximilian. "Quantifying economic damages from climate change." Journal of Economic Perspectives 32, no. 4 (2018): 33-52.

Barreca, Alan, Karen Clay, Olivier Deschenes, Michael Greenstone, and Joseph S. Shapiro. "Adapting to climate change: The remarkable decline in the US temperature-mortality relationship over the twentieth century." Journal of Political Economy 124, no. 1 (2016): 105-159.
Carleton, Tamma, Amir Jina, Michael Delgado, Michael Greenstone, Trevor Houser, Solomon Hsiang, Andrew Hultgren et al. "Valuing the global mortality consequences of climate change accounting for adaptation costs and benefits." The Quarterly Journal of Economics 137, no. 4 (2022): 2037-2105.

Deschênes, Olivier, and Michael Greenstone. "Climate change, mortality, and adaptation: Evidence from annual fluctuations in weather in the US." American Economic Journal: Applied Economics 3, no. 4 (2011): 152-185.

Heutel, Garth, Nolan H. Miller, and David Molitor. "Adaptation and the mortality effects of temperature across US climate regions." The review of economics and statistics 103, no. 4 (2021): 740-753.

Rode, Ashwin, Tamma Carleton, Michael Delgado, Michael Greenstone, Trevor Houser, Solomon Hsiang, Andrew Hultgren et al. "Estimating a social cost of carbon for global energy consumption." Nature 598, no. 7880 (2021): 308-314.

Wenz, Leonie, Anders Levermann, and Maximilian Auffhammer. "North–south polarization of European electricity consumption under future warming." Proceedings of the National Academy of Sciences 114, no. 38 (2017): E7910-E7918.

---

## Referee Report (RR1)

Thank you for responding to my comments and questions regarding your manuscript. After reading the updated manuscript and response to reviewers, I am convinced that the article does not need any meaningful additional analyses. The calculations conducted by the authors represent an important and comprehensive assessment of sector-specific climate damages in the United States and will undoubtedly have profound policy impact.

However, I am not fully satisfied with the authors' replies to my concerns. I will note that I found the response very difficult to digest because no precise textual changes were listed and no text from the manuscript was reproduced, so I cannot directly trace what has been changed in response to each of my concerns. I aim to provide an accurate reflection of manuscript updates below, but it is possible I have missed something because no direct links to the updated text were provided.

Second, the authors largely responded directly to me regarding each of the limitations of the analysis that I had raised. But in most cases, the text of the manuscript did not change in a meaningful way. I am convinced that the authors have an important contribution without needing to address each of these limitations head on. However, I do think it is necessary for key assumptions and limitations to be transparently communicated to the readers. I list the specific cases where I think this is critical below.

1. Adaptation. The authors insufficiently communicate that most sectors in their analysis assume no adaptation takes place (see Table A2-1). Text on lines 198-204 mentions that different adaptation scenarios are modeled to the extent possible given available scientific evidence, but this text should also explicitly note that only two of 20 sectors include proactive adaptation, making it likely that estimated impacts are overly severe. Given that temperature-related mortality dominates their overall damage estimates, the fact that these results ignore adaptation, despite strong evidence of historical adaptation (Heutel et al., 2021; Barreca et al., 2016), should be openly discussed and justified.

2. Income effects on the damage function. The authors state that "most of the sectoral damages are proportional to GDP per capita" (line ~275). In our exchange, we agree that in some cases damages may be higher in wealthier populations, but in other cases the reverse may be true. Instead of omitting this important point, the authors should mention that they make this structural assumption despite evidence that income could push damages in either direction.

3. Spatial heterogeneity of warming. The authors should clearly state that they ignore uncertainty in the spatial distribution of warming across the United States. It makes sense to cite Sarofim et al. (2021) to justify this choice, but it shouldn't be left to the reader to figure out what forms of climate uncertainty are or are not included.

4. Damage function uncertainty. The authors show some uncertainty in damage functions for mortality by comparing *across* impact models. However, many impact models, in particular econometric ones, also have damage function uncertainty *within* each model. The authors should clearly state that such damage function uncertainty is omitted. Showing that the range across these three modeling approaches lies within the

distribution of climate uncertainty results does not address my concern – if you combined damage function uncertainty with climate model uncertainty, as is recommended (Burke et al., 2015), results would undoubtedly convey far larger uncertainty ranges.

5.  Electricity. The authors reply to me, oddly, by stating that although they include only electricity, the study they draw on also has natural gas. Why not include natural gas, then? At a minimum, the authors should clearly convey that natural gas expenditures are not included in their analysis (noting that their inclusion would likely lower damages).

6.  Overlapping sectors and sector interlinkages. We can agree to disagree on the importance of these issues in the current FrEDI results, but the authors should at least mention the limitations of FrEDI in these areas. Its omission from the manuscript, given the many-sector bottom-up approach, is troubling.

7.  I remain confused by lines 85-90, which imply that a user could pair *any* climate scenario with *any* socioeconomic scenario, which is not what the authors do in this paper (and which is concerning, given logical inconsistencies that could arise).

8.  The authors stated that they included a footnote clarifying that their calculations are equivalent to a domestic SCC, but I cannot find that footnote anywhere.

---

## Author Response (AR2)

**Response to Reviewers for Hartin et al., Advancing the estimation of future climate impacts within the United States**

*We thank the reviewers for providing additional comments and feedback on the manuscript. These have helped us further refine sections where additional clarification and discussion in the main text were required, such as further details on the treatment of adaptation within the FrEDI framework and clarification regarding the types of uncertainty accounted for in the main and supplemental analyses and the requested changes did not result in new or additional analyses. We have responded to each specific comment below in italics and have updated the text accordingly. The line numbers associated with our responses correspond to the line numbers in the clean version of the resubmitted text document.*

**Reviewer #1**

The manuscript is moving in a positive direction. I have a few remaining concerns, almost entirely related to clearly communicating the caveats and limitations and outline them below.

Caveats – I feel that there remains a need for stronger caveats to make clear the assumptions and approximations that must be made to carry out this kind of analysis. I first want to recognize that what constitutes the "best available science" depends on the spatial and temporal scales we're talking about and requires balancing representing more processes against more uncertainties. Stylized relationships are a requirement when doing this kind of impacts work on the national or global scale, but of course we don't want to oversimplify or neglect key processes. Even experts will disagree about what constitutes "key processes" (deep uncertainty). I'll note below a few places where I feel stronger justification/caveats are needed in light of the inherent uncertainties, but in my view this further emphasizes the need for tools (like FrEDI) to explore uncertainties.

*We thank the reviewer for the additional comments and have addressed each specific concern below. We also agree that the phase, "best available science", is dependent upon the applicable temporal and spatial scales, as well as uncertainty characterization and thresholds. This phrase does not appear in the manuscript or its supplement. Please see below for our specific responses and changes.*

Line 114 - multi-century projections – certainly socioeconomic, but climate as well – have a lot of asterisks' involved. It's important to be up front about the limitations too. For example, the lack of tipping points could be quite relevant in high-emissions scenarios wherein Greenland melt could slow the AMOC and greatly cool the northern hemisphere. Same goes for potential runaway Antarctic melt. I would characterize these limitations as structural uncertainties that FrEDI is ideally suited to characterize by its modular nature. I see the note about tipping points at Line 415 - can you expand this discussion to justify that even without such mechanisms, the results are still reliable and useful?

*We agree with the reviewer that there are large and important uncertainties in multi-century projections and analyses, and that under some of the very high emission scenarios, tipping points may be reached within the Earth system. To further clarify these uncertainties in the input scenarios and resulting analysis, we have added the following text to the methods and discussion sections.*

*Line 125. While uncertainties multi-century projections are considerable, as discussed in Rennert et al., 2021, these projections represent the largest set of probabilistic socioeconomic and emissions scenarios based on high-quality data, robust statistical techniques, and expert elicitation.*

*Line 136. While FaIR only captures uncertainties in those feedbacks and climate tipping points that are apparent in more sophisticated Earth system models or the historic record to which FaIR is calibrated, FaIR does include uncertainties in parameters such as the equilibrium climate sensitivity, transient climate response, present-day aerosol radiative forcing, present-day $CO_2$ concentrations, and recent-past ocean heat content change.*

*Line 449. We recognize that multi-century projections are inherently challenging. This is particularly true for socioeconomic projections of GDP, population, and technologies: even projections to the end of the century have been challenged (Barron, 2018). The climate system is better understood, but FaIR only captures the effects of those feedbacks and tipping points which are apparent in the GCMs and historic record to which FaIR was calibrated.*

*Line 488. Future work may entail coupling BRICK to the framework to better explore the uncertainty within sea level rise (Wong et al., 2022, 2017) or coupling to an alternative reduced-form climate model, Hector, to explore permafrost thaw (Woodard et al., 2021). Without explicit representation of some of these feedbacks, we can view these results as potentially lower bound damage estimates.*

Paragraph at Line 149 – What assumptions about migration are baked into the regional population projections? Somewhat related to what I recall the other reviewer noted, there are a number of likely interrelated/overlapping structural uncertainties in here - for example, assumptions about migration and retreat from inundated coastal areas.

*The national-level population projections are from the RFF-SPs. At the national level, the RFF-SP population scenarios are built by forward projecting changes in fertility, mortality, and international migration, by age and sex and country, as described in Rennert et al., 2021. In the RFF-SPs, these probabilistic projections of net international migration did not consider future climate changes, such as climate induced migration. At the regional scale, as described in the main text, relative populations in each of the 7 FrEDI regions were taken from years 2010-2090 in EPA's ICLUS model. Like the RFF-SPs, these projections do not account for climate driven migration within the U.S. We have adjusted the text in the Methods section to clarify this detail.*

Line 193 – Is this using census data from 2014-2018, held constant, to estimate future demographic breakdowns in population out to 2090, or 2300? Stronger justification/caveats are needed here too. Can you refer in the main text to specific text/sections in appendix for details about social vulnerability calculations/assumptions? It's still a bit unclear in the main text how this is done, and a clearer picture will help put the assumptions/limitations into context.

*To clarify how we use U.S. census data for estimating differential climate-driven impacts across different populations, we have moved the following text from the Appendix up into the Methods section. As described in this section, total U.S. and regional populations change over time (driven by ICLUS data), but the relative percentages of each population group within each Census tract are held constant over time. These relative population counts are help constant because long-term projected changes in regional demographic patterns are not available.*

*Line 215: These differential impacts are calculated in FrEDI at the Census tract level as a function of current population demographic patterns (i.e., percent of each group living in each census tract) (U.S. Census), projections of CONUS population (U.S. EPA, 2017), and projections of where climate-driven*

*damages are projected to occur (from Census tract-level temperature-impact relationships in FrEDI). The relative percent of each group in each Census tract is from the 2014-2018 U.S. Census American Community Survey dataset (U.S. Census) and is held constant over time because robust and long-term projections for local changes in demographics out to 2090 and beyond are not readily available. We consider four categories for which there is evidence of differential vulnerability (Table A2), including low income, ethnicity, and race[1], educational attainment, and age.*

I appreciate the extended discussion in Sec. A3 about adaptation assumptions. Can this be called out specifically in the main text? For example, just a sentence or two that acknowledges that this sort of climate impacts modeling is important, but necessitates assumptions and approximations. You could mention one or two prominent ones in main text (e.g., no proactive relocation in light of coastal risks), then refer the reader to A3 for further discussion. Relatedly, I note that many contemporary large-scale studies of coastal adaptation and impacts are similarly limited in a lack of proactive retreat in the face of sea-level rise and increasing storm surge hazards. There are some that include it, though they're in the minority.

*At the reviewer's request, we have moved the following text to the methods and results sections to further clarify our treatment of adaptation and the associated sensitivity of our results. We also added further detail on the treatment of adaptation in the temperature-related mortality sector, in response to additional comments from Reviewer 2.*

*Line 196. As discussed further in Section A3, Reactive or Reasonably Anticipated Adaptation is where decision makers respond to climate change impacts by repairing damaged infrastructure (e.g., road or rail repair) or reactively responding to current conditions (e.g., building sea walls or beach nourishment), but do not take actions to prevent or mitigate future climate change impacts. No Additional Adaptation largely incorporates historical or current levels of adaptive mitigation that were in place during the time period of each underlying sectoral study. Example sensitivities to projected climate-driven damages are explored within section 3.1 and A3.*

*Line 337. These sectoral damages are sensitive to assumptions in the adaptation scenarios (see section A3 for more detail). For example, the coastal property sector considers three different adaptation options, no adaptation, reactive, and proactive adaptation. The underlying model within this sector, the National Coastal Property Model, has options for optimal ("proactive") response to future sea level rise, "reactive" or reasonably anticipated response to current conditions (including sea walls, beach nourishment, house elevation, or managed retreat), or rebuilding in place as often as necessary. Historical data suggests that most of our response to sea level rise thus far is in between reactive adaptation and no adaptation (Lorie et al., 2020). Considering the range of possible adaptation options in this coastal property sector, mean damages range from $17 billion USD under no adaptation to $7.5 billion USD under proactive adaptation. Damages under the default 'reactive' adaptation assumption are $9.4 billion USD. While the inclusion of adaptation options for any sector within FrEDI depends on the consideration and treatment of adaptation in the underlying impact studies, Table A3 further illustrates that projected climate-driven damages are sensitive to adaptation options in each sector where they are considered. Notably, the largest impact sector in this study, temperature-related mortality does not*

*include assumptions about future adaptation. While the primary underlying study (Cromar et al., 2022) is a well-regarded meta-analysis of existing global temperature-related mortality studies, it does not explicitly consider future adaptive measures. Exploring projected 2090 damages from one alternative damage function that assesses impacts of extreme temperature on mortality in 49 U.S. cities (Mills et al., 2014), suggests that damages will be significantly reduced (Table A4) in the event that U.S. cities can gradually adapt to hotter temperatures, for example through physical acclimatation, increased air conditioning penetration, and behavioral changes. Several other studies have also observed reductions in temperature-related vulnerability over time (Lay et al., 2021), however there is little consensus regarding the most appropriate way to consider future adaptation in this sector, even though several methods have been applied (Sarofim, M.C. et al., 2016; Carleton et al., 2022; Heutel et al., 2021). Therefore, we use the most recently published meta-analysis for the central estimate in this analysis, but also present results from alternative assumptions and studies (Tables A3 and A4), further illustrating the unique advantage of the FrEDI framework of enabling direct comparisons across studies.*

Line 99 – Can you give more details about what specifically from AR6 or other data the FAIR simulations were calibrated with, and in what ways this addresses the potential issues of incompatible SSP-RCP or other socioecomic-emissions/temperature pathways? I get the sense that they're internally consistent because temperatures and populations (eg) are both generated within the MimiGIVE model, or something along these lines. Some specificity or further details here would clear that up.

*We have added the following text in the Methods section to provide more specifics about the parameters with uncertainty distributions that are included within FaIR. We also note that the FaIR uncertainty parameter set was not only calibrated to the IPCC AR6 assessment of present day warming, but the IPCC then used FaIR with this calibrated parameter set for their projections of future warming probabilities in the AR6 report.*

*Line 102 - The FaIR calibration is consistent with the IPCC AR6 Working Group 1 assessment of present-day warming, equilibrium climate sensitivity, transient climate response, present-day aerosol radiative forcing, present-day $CO_2$ concentrations, and recent-past ocean heat content change, including the uncertainties in these distributions (Forster et al. 2021; Smith et al. 2021).*

*Line 136. While FaIR only captures uncertainties in those feedbacks and climate tipping points that are apparent in more sophisticated Earth system models or the historic record to which FaIR is calibrated, FaIR does include uncertainties in parameters such as the equilibrium climate sensitivity, transient climate response, present-day aerosol radiative forcing, present-day $CO_2$ concentrations, and recent-past ocean heat content change.*

*We also note that the RFF-SP emission and socioeconomic projections were developed to be internally consistent. Following the approach described in Rennert et al., 2022, we then randomly sample emission timeseries from the RFF-SPs and FaIR parameter sets from the calibrated population (2237 calibrated sets out of a possible 1 million). As reviewer 2 also had similar clarifying questions regarding these details and potential limitations, we now also clarify in the Methods section that this approach of separately treating emission and climate uncertainties does not allow us to account for potential feedbacks of certain climate parameters on, for example, changes in climate-driven damages and the resulting impacts on U.S. GDP/population.*

Line 109. *However, there remain some limitations in that separately considering climate parameter and socioeconomic uncertainty ignores potential feedbacks from observed climate change onto socioeconomics (e.g., a higher climate sensitivity could result in larger climate-driven damages, which could lead to lower emissions or GDP than would occur in a lower climate sensitivity world).*

**Reviewer #2**

Thank you for responding to my comments and questions regarding your manuscript. After reading the updated manuscript and response to reviewers, I am convinced that the article does not need any meaningful additional analyses. The calculations conducted by the authors represent an important and comprehensive assessment of sector-specific climate damages in the United States and will undoubtedly have profound policy impact. However, I am not fully satisfied with the authors' replies to my concerns. I will note that I found the response very difficult to digest because no precise textual changes were listed and no text from the manuscript was reproduced, so I cannot directly trace what has been changed in response to each of my concerns. I aim to provide an accurate reflection of manuscript updates below, but it is possible I have missed something because no direct links to the updated text were provided. Second, the authors largely responded directly to me regarding each of the limitations of the analysis that I had raised. But in most cases, the text of the manuscript did not change in a meaningful way. I am convinced that the authors have an important contribution without needing to address each of these limitations head on. However, I do think it is necessary for key assumptions and limitations to be transparently communicated to the readers. I list the specific cases where I think this is critical below.

*We thank the reviewer for their careful re-review of this manuscript. It has been greatly improved as a result. We have responded to each comment below and provided explicit excerpts of additions or changes we made to the text in response.*

1. Adaptation. The authors insufficiently communicate that most sectors in their analysis assume no adaptation takes place (see Table A2-1). Text on lines 198-204 mentions that different adaptation scenarios are modeled to the extent possible given available scientific evidence, but this text should also explicitly note that only two of 20 sectors include proactive adaptation, making it likely that estimated impacts are overly severe. Given that temperature-related mortality dominates their overall damage estimates, the fact that these results ignore adaptation, despite strong evidence of historical adaptation (Heutel et al., 2021; Barreca et al., 2016), should be openly discussed and justified.

*Thank you for this comment. Reviewer 1 also requested that we add more explicit discussion of adaptation to the main text. In response, we moved some of our discussion of adaptation from the Appendix to the main text. We have also added a new section discussing the treatment of adaptation in response to the reviewers point about adaptation in the temperature-related mortality sector.*

*Line 338. These sectoral damages are sensitive to assumptions in the adaptation scenarios (see section A3 for more detail). For example, the coastal property sector considers three different adaptation options, no adaptation, reactive, and proactive adaptation. The underlying model within this sector, the National Coastal Property Model, has options for optimal ("proactive") response to future sea level rise, "reactive" or reasonably anticipated response to current conditions (including sea walls, beach nourishment, house elevation, or managed retreat), or rebuilding in place as often as necessary.*

*Historical data suggests that most of our response to sea level rise thus far is in between reactive adaptation and no adaptation (Lorie et al., 2020). Considering the range of possible adaptation options in this coastal property sector, mean damages range from $17 billion USD under no adaptation to $7.5 billion USD under proactive adaptation. Damages under the default 'reactive' adaptation assumption are $9.4 billion USD. While the inclusion of adaptation options for any sector within FrEDI depends on the consideration and treatment of adaptation in the underlying impact studies, Table A3 further illustrates that projected climate-driven damages are sensitive to adaptation options in each sector where they are considered. Notably, the largest impact sector in this study, temperature-related mortality does not include assumptions about future adaptation. While the primary underlying study (Cromar et al., 2022) is a well-regarded meta-analysis of existing global temperature-related mortality studies, it does not explicitly consider future adaptive measures. Exploring projected 2090 damages from one alternative damage function that assesses impacts of extreme temperature on mortality in 49 U.S. cities (Mills et al., 2014), suggests that damages will be significantly reduced (Table A4) in the event that U.S. cities can gradually adapt to hotter temperatures, for example through physical acclimatation, increased air conditioning penetration, and behavioral changes. Several other studies have also observed reductions in temperature-related vulnerability over time (Lay et al., 2021), however there is little consensus regarding the most appropriate way to consider future adaptation in this sector, even though several methods have been applied (Sarofim, M.C. et al., 2016; Carleton et al., 2022; Heutel et al., 2021). Therefore, we use the most recently published meta-analysis for the central estimate in this analysis, but also present results from alternative assumptions and studies (Tables A3 and A4), further illustrating the unique advantage of the FrEDI framework of enabling direct comparisons across studies.*

2. Income effects on the damage function. The authors state that "most of the sectoral damages are proportional to GDP per capita" (line ~275). In our exchange, we agree that in some cases damages may be higher in wealthier populations, but in other cases the reverse may be true. Instead of omitting this important point, the authors should mention that they make this structural assumption despite evidence that income could push damages in either direction.

*As shown in Table A5 and discussed further in A4, FrEDI uses at least one type of socioeconomic scaling factor to monetize climate-driven damages across 14 of the sectors (or variants within sectors) currently included within the framework. These scale factors are derived from the same underlying studies from which the damage functions have been derived, which generally show damages increasing with GDP per capita. For example, the number of mortality cases in the health-related sectors are proportional to population, and the valuation of these cases scales with GDP per capita, such that the willingness to pay to reduce fatality risk (i.e., VSL) is adjusted based on the projection of GDP per capita and an income elasticity of 1. In other sectors where wealthier populations may increase the potential for various forms of adaptation to reduce damages (e.g., increased air conditioning to reduce heat mortality), these actions also come at a cost, and where available in the underlying studies, FrEDI provides the ability to explore these adaptation scenarios. We have added the following text to the original sentence to clarify that this structural assumption reflects the data in the underlying sector studies currently within the framework and have added additional details about the treatment of adaptation (see response to reviewer comment #1)*

*Line 263. Because most of the sectoral damages as determined from the underlying sectoral models are proportional to GDP per capita (given that the default elasticity of VSL to GDP per capita is 1, all sectors*

*with a mortality endpoint also qualify), a correction can be made to account for this relationship (Nordhaus, 2017).*

3. Spatial heterogeneity of warming. The authors should clearly state that they ignore uncertainty in the spatial distribution of warming across the United States. It makes sense to cite Sarofim et al. (2021) to justify this choice, but it shouldn't be left to the reader to figure out what forms of climate uncertainty are or are not included.

*We have included the following text within the Methods section to clarify our treatment of regional warming and relevant uncertainties. While FrEDI does provide the option to explore differences in regional warming that are associated with spatial warming heterogeneity in the 6 GCMs also used in the underlying studies (e.g., CanESM2, CCSM4, GFDL-CM3, GISS-E2-R, HadGEM2-ES, MIROC5), results presented in this analysis reflect the average across the GCM ensemble.*

*Line 182. Sub-national differences in warming are also explored within FrEDI using results derived from a consistent set of GCMs that were also used within the underlying studies (e.g., Sarofim et al., 2021). For example, unique damage functions for each sector (and variant within each sector) are developed for each region and GCM, based on its relationship to CONUS temperature. While FrEDI outputs damages by region and GCM, the main results in this analysis present national and regional damages calculated from the average across the GCM ensemble.*

4. Damage function uncertainty. The authors show some uncertainty in damage functions for mortality by comparing across impact models. However, many impact models, in particular econometric ones, also have damage function uncertainty within each model. The authors should clearly state that such damage function uncertainty is omitted. Showing that the range across these three modeling approaches lies within the distribution of climate uncertainty results does not address my concern – if you combined damage function uncertainty with climate model uncertainty, as is recommended (Burke et al., 2015), results would undoubtedly convey far larger uncertainty ranges.

*Very few studies explore uncertainty within the damage function space. Two of the temperature-related mortality sectors within FrEDI do provide information that we can use to develop two additional damage functions that reflect the 90% confidence interval associated with the parametric damage function uncertainty in each underlying study. We present impacts are predicted across these uncertainty intervals for each study, as calculated from the mean warming and socioeconomic scenarios from the RFF-SPs. We use the mean RFF-SP scenario in order to be able to more directly compare the different levels of uncertainty associated with climate and socioeconomics compared to the uncertainties associated with the damage functions. We have added the following text in the Results and Discussion section to clarify the sources of uncertainty that are included in our main text results. We also added information in the Appendix on this supplemental temperature-related mortality damage function analysis, as well as information about the combined damage function and climate uncertainties.*

*Line 283 - Confidence intervals presented throughout this section include uncertainty in GDP, population, and climate parameters, but do not account for additional sectoral parametric or structural uncertainty.*

*Line 412. In addition to these uncertainties and sensitivities to adaptation options, damage estimates within FrEDI are also sensitive to uncertainties in the underlying damage functions themselves. Similar to adaptation, FrEDI can incorporate parametric uncertainty in each damage function when the relevant*

*information is available in the underlying study, as well as this source of structural uncertainty when uncertainty estimates are available in the underlying study or when multiple damages functions are available for a single sector. For example, as further described in section A4, FrEDI incorporates three studies of climate-driven temperature-related mortality, two of which include underlying uncertainty estimates.*

*Line 598. The Cromar et al., 2022 study also provides a standard error on the impact function relative risk coefficient, which was used to develop a 90% confidence interval around this parameter. The 90% confidence interval supports the calculation of impacts for the low and high end of the confidence interval (5th and 95th percentile values) within FrEDI, as well as a central estimate which corresponds to the mean result. The Hsiang et al., 2017 study authors also shared results from uncertainty modeling in the underlying work, which was also used to develop a 90% confidence interval of results. These uncertainty results support the calculation of the low and high end of the confidence interval (5th and 95th percentile values) within FrEDI, as well as a central estimate which corresponds to the median result (50th percentile).*

*There are currently three underlying temperature-related mortality studies within FrEDI. Table A4 provides a snapshot of the parametric uncertainty within each temperature-related mortality estimate, as well as structural damage function uncertainty by comparing impacts across multiple studies. To separately evaluate the level of damage-function-related uncertainty compared to other sources of uncertainty presented in the main text (e.g., socioeconomics & climate), we show the mean damages from each damage function in Table A4, as calculated as the average across the RFF-SPs, as well as the $90^{th}$ confidence intervals, as calculated by taking the average across the RFF-SPs for the damages predicted by the high and low confidence interval damage functions. Compared to Table A1, Table A4 shows smaller predicted ranges in temperature-related mortality damages than the ranges in damages derived from combined uncertainties in socioeconomic and climate parameters. We do not present these uncertainty levels in the main text as only a one sector currently included with the FrEDI framework include information that allow us to evaluate parametric and structural damage function uncertainty.*

5. Electricity. The authors reply to me, oddly, by stating that although they include only electricity, the study they draw on also has natural gas. Why not include natural gas, then? At a minimum, the authors should clearly convey that natural gas expenditures are not included in their analysis (noting that their inclusion would likely lower damages).

*Thank you for clarifying this point. As noted this is not a comprehensive accounting of all impacts to the U.S. and we continue to update the tool. We will explore whether we can include expenditures from natural gas within our framework.*

6. Overlapping sectors and sector interlinkages. We can agree to disagree on the importance of these issues in the current FrEDI results, but the authors should at least mention the limitations of FrEDI in these areas. Its omission from the manuscript, given the many-sector bottom-up approach, is troubling.

*We have added an additional paragraph to the Results and Discussion section in the main text to better describe these details of overlapping sectoral damages within FrEDI.*

*Line 363. The sectors assessed in this study are independent and therefore damages are additive across these sectors. One potential exception could be temperature-related mortality and the climate-air quality*

*linkage, as most approaches to estimating temperature-related mortality are statistical rather than mechanistic, which could lead to double counting of some health effects between these two sectors. Specifically, (Cromar et al., 2022) note that it will be important to continue exploring potential synergies between the effects of temperature and air pollution to adequately capture the potential risk in compound climate events such as these. Conversely, there can also be compounding effects that the FrEDI analytical approach does not account for: e.g., power outages due to increased summer electricity demand could exacerbate temperature-related mortality. However, few studies produce quantitative, monetized estimates of compounding or interacting effects at the national scale as would be required to build into comprehensive impact tools (Clarke et al. 2018).*

7. I remain confused by lines 85-90, which imply that a user could pair any climate scenario with any socioeconomic scenario, which is not what the authors do in this paper (and which is concerning, given logical inconsistencies that could arise).

*As the reviewer notes, we use a Monte Carlo approach to couple FaIR climate model parameter sets (that account for uncertainties in climate model parameters) with paired emission/socioeconomic projections that were previously developed by RFF. This is the same sampling approach that was recently employed by Rennert et al., 2022. We have adjusted the text in multiple locations in the introduction and methods sections to clarify that the emissions (and resulting temperature) and socioeconomic projections used in this analysis are internally consistent. This approach of pairing a specific socioeconomic scenario with any climate \*parameter\* scenario is equivalent to pairing a single SSP scenario (e.g., SSP2-4.5) with different GCMs (which embody different climate parameter sets). However, this approach does have limitations and (also in response to a comment from reviewer 1), we have added additional text in the Methods section noting that this sampling approach cannot account for feedbacks between climate change and socioeconomics.*

*Line 76. In this study, we use 10,000 recently developed, paired probabilistic emissions and socioeconomic projections, in combination with resulting temperature projections from a simple climate model as inputs to FrEDI, which is then run to quantify the annual physical and economic impacts associated with each resulting paired climate and socioeconomic scenario through the end of the 21$^{st}$ century across the contiguous United States (CONUS).*

*Line 91. First, projections of global greenhouse gas emissions (Figure 1, Input 1) are used as input to a simple climate model to derive trajectories of changes in global mean surface temperature (Figure 1, Output 1). These emission projections were developed as paired scenarios with projections of national-level population and GDP, and therefore the resulting temperature trajectories from the simple climate model are then passed to FrEDI (Figure 1, Input 2) alongside the paired projections of U.S. Population and GDP (Figure 1, Input 1) to model annual long-term climate damages across 20 impact sectors, seven CONUS regions, multiple adaptation scenarios, and socially vulnerable populations (Figure 1, Output 2).*

*Line 109. However, there remain some limitations in that separately considering climate parameter and socioeconomic uncertainty ignores potential feedbacks from observed climate change onto socioeconomics (e.g., a higher climate sensitivity could result in larger climate-driven damages, which could lead to lower emissions or GDP than would occur in a lower climate sensitivity world).*

8. The authors stated that they included a footnote clarifying that their calculations are equivalent to a domestic SCC, but I cannot find that footnote anywhere.

*Thank you for catching this mistake. We added the footnote back in on page 18.*

*Footnote 10 Net present damages resulting from an additional ton of $CO_2$ emissions is sometimes characterized as a "domestic social cost of carbon".*